# PARTAFFORD: PART-LEVEL AFFORDANCE DISCOVERY

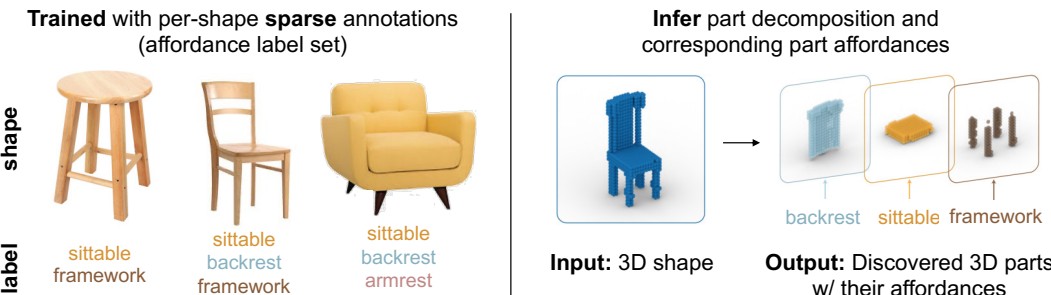

Figure 1: **The proposed PartAfford: discover 3D object part affordances by learning contrast in affordance compositions.** During training (**left**), given weak annotations (per-shape affordance label set), a learning framework is devised to ground affordance (*e.g.*, backrest) to 3D part (*e.g.*, sofa back) through learning cross-category, affordance-related shapes (*e.g.*, chair, sofa) with various affordance compositions. At test time (**right**), the learned model decomposes the 3D object into parts and infers the part-level affordances.

## ABSTRACT

Understanding what objects could furnish for humans—learning object *affordance*—is the crux of bridging perception and action. In the vision community, prior work has primarily focused on learning object affordance with dense (*e.g.*, at a per-pixel level) supervision. In stark contrast, we humans learn the object affordance *without* dense labels. As such, the fundamental question to devise a computational model is: What is the natural way to learn the object affordance from geometry with humanlike weak supervision? In this work, we present the new task of part-level affordance discovery (PartAfford): Given only the affordance labels for each object, the machine is tasked to (i) decompose 3D shapes into parts and (ii) discover how each part of the object corresponds to a certain affordance category. We propose a novel learning framework that discovers part-level representations by leveraging only the affordance set supervision and geometric primitive regularization *without* dense supervision. To learn and evaluate PartAfford, we construct a part-level, cross-category 3D object affordance dataset, annotated with $24$ affordance categories shared among $> 25,000$ objects. We demonstrate through extensive experiments that our method enables both the abstraction of 3D objects and part-level affordance discovery, with generalizability to difficult and cross-category examples. Further ablations reveal the contribution of each component.

## 1 INTRODUCTION

The human vision system could swiftly locate the functional part upon using an object for specific tasks (Land et al., 1999). Such a critical capability in object interaction requires fine-grained object *affordance* understanding. *Affordance*, coined and originally theorized by Gibson (Gibson & Carmichael, 1966; Gibson, 1979), characterizes how humans interact with human-made objects and environments. As such, affordance understanding of objects and scenes has a significant influence on bridging visual perception and holistic scene understanding (Huang et al., 2018b;a; Chen et al., 2019) with actionable information (Soatto, 2013; Han et al., 2022).

Object affordances have two main characteristics. First, object affordances are not defined in terms of conventional categorical labels in computer vision; instead, they are defined by the associated actions

for various tasks and are naturally *cross-category*. For example, both chair and sofa can be sat on, which indicates they share the *sittable* affordance. Similarly, desktop and bookshelf share the *support* affordance. Second, object affordances are intrinsically *part-based*. We could easily associate *sittable* affordance with the seats of chairs and sofas, and *support* with the boards of desktop and bookshelf. As such, the ability to learn **part-based**, **cross-category** affordance is essential to demonstrate the general object affordance understanding.

In passive affordance learning, prior literature follows the supervised learning paradigm, in which dense affordance annotation on the objects is fed as supervised signals (Deng et al., 2021). However, this line of thought depends heavily on the quality of dense annotation, which significantly deviates from how we humans learn to understand affordance. Humanlike supervision would be: "you can sit on this chair and rest your arm," "you can open the lid and hold water with the cup." In this paper, we try to answer: How to distinguish each object part while recognizing corresponding affordances with such weak and natural supervisions?

To tackle this problem, we present *PartAfford*, a new task of part-level affordance discovery, which learns the object affordance with the natural supervision of the affordance set. As shown in Fig. 1, by providing only the set of affordance labels for each object, the algorithm is tasked to decompose the 3D shapes into parts and discover how each part corresponds to a certain affordance category, which is challenging and under-explored in the area of generalizable part-level object understanding and affordance learning.

To address this, we propose a novel method that discovers part-level representations with self-supervised 3D reconstruction, affordance set supervision, and primitive regularization. The proposed approach consists of two main components. The first component is an encoder with slot attention for unsupervised clustering and abstraction. Specifically, we encode the 3D object into visual features and abstract the low-level features into a set of *slot* variables (Locatello et al., 2020). The second component is a decoder built upon the learned slot features. It has three output branches that jointly reconstruct the 3D parts and object, predict the affordance labels, and regularize the learned part-level shapes with cuboidal primitives. Our method does not rely on dense supervision but instead learns from the weak set supervision. It discovers the part-level affordance by learning the correspondence between affordance labels and abstracted 3D object parts.

Learning and evaluating *PartAfford* demands collections of 3D objects and their affordance labels for object parts. Prior work on visual affordance learning (Hassanin et al., 2021) either focuses on 2D objects and scenes or lacks part-based annotation (Deng et al., 2021). Hence, we construct a part-level, cross-category 3D object affordance dataset annotated with 24 affordance categories shared among over 25,000 3D objects. The 3D objects are collected from PartNet dataset (Mo et al., 2019b) and the PartNet-Mobility dataset (Xiang et al., 2020). The 24 part affordance categories are defined in terms of adjectives (*e.g.*, "sittable") or nouns (*e.g.*, "armrest"); they describe how object parts could afford human daily actions and activities. We annotate the part-level object affordances by manually mapping the fine-grained object part defined in PartNet to the part affordances defined in this work.

By experimenting on this newly constructed *PartAfford* dataset, we empirically demonstrate that our method jointly enables the abstraction of 3D objects and part-level affordance discovery. Our model also shows strong generalizability on hard and cross-category objects. Further experiments and ablations analyze each component's contribution and point out future directions.

In summary, our work makes four main contributions:

- We present a new *PartAfford* task for part-level affordance discovery. Compared to the prior densely-supervised learning paradigm, *PartAfford* learns the visual object affordance more naturally.
- We propose a novel learning framework for tackling *PartAfford*, which jointly abstracts 3D objects into part-level representations and discovers affordances by learning the affordance correspondence.
- We build the benchmark for learning and evaluating *PartAfford* by curating a dataset consisting of 3D objects and annotating part-level affordances.
- We empirically demonstrate the efficacy and generalization capability of the proposed method and analyze each component's significance via a suite of ablation studies. Code and data will be released for research purposes.

## 2 RELATED WORK

**Affordance Learning**   Affordance learning is a multidisciplinary research field of vision, cognition, and robotics. In general, "affordance" is first perceived from images (Gupta et al., 2011; Kjellström et al., 2011; Zhu et al., 2015; Myers et al., 2015; Roy & Todorovic, 2016) or videos (Xie et al., 2013; Zhu et al., 2016; Fang et al., 2018; Nagarajan et al., 2020), followed by cognitive reasoning (Zhu et al., 2015; 2020), and finally serves for task and motion planning in robotics (Nagarajan & Grauman, 2020; Mo et al., 2022). Prior work tackles affordance at various scales and representations. Although affordance has been studied at the scene level (Zhao & Zhu, 2013; Gupta et al., 2015; Roy & Todorovic, 2016), object level (Nguyen et al., 2017; Mo et al., 2021; Gadre et al., 2021), and associated with generated human poses (Zhu et al., 2015; Wang et al., 2017b), few attempts study affordance as a 3D shape analysis task (Yu et al., 2015; Liang et al., 2016; Zhu et al., 2016; Wang et al., 2017a) since it would normally require large-scale, high-quality 3D data. A recent work (Deng et al., 2021) benchmarks several affordance estimation tasks on PartNet (Mo et al., 2019b) with dense affordance heatmap supervisions, annotated by densely selecting keypoints without considering affordance compositionality. Prior work also tackles affordance learning through interaction-based methods, either from human demonstration videos (Kjellström et al., 2011; Nagarajan et al., 2019; 2020) or simulation-based active learning (Wang et al., 2022; Mo et al., 2021). The first usually infers high-level and coarse 2D affordance, and the second is often restricted to basic manipulations in specific domains. In comparison, *PartAfford* studies affordance in a weakly supervised manner, such that the affordance discovery will be guided by affordance set matching and geometry abstraction. The new affordance dataset we construct provides fine-grained, part-level 3D affordance annotations, tailored for the weak supervision setting and affordance compositionality study.

**Object-centric Learning**   Object discovery has been studied in an iterative end-to-end fashion (Greff et al., 2017; Van Steenkiste et al., 2018; Burgess et al., 2019; Engelcke et al., 2019; Greff et al., 2019; Du et al., 2021). Recently, Locatello et al. (2020) presents the slot attention module, an efficient and generic framework for object-centric representation extraction. It is capable of modeling compositional nature in synthetic scenes with multiple simple geometry shapes (Kabra et al., 2019). Subsequently, Stelzner et al. (2021) and Yu et al. (2021) apply slot attention on unsupervised 3D-aware scene decomposition, integrating NeRF (Mildenhall et al., 2020) as object representations. They demonstrate that slot-based bottleneck could perform reasonably on synthetic multi-view RGB datasets with a textureless background. In the weakly-supervised regime, methods have been proposed for 3D semantic segmentation with scene-level labels (Wei et al., 2020; Ren et al., 2021). However, these methods rely on additional input (*e.g.*, color, normal) and abstract object-level features in different data domains. Our work takes one step further to tackle the challenge of *part-level* affordance discovery of 3D objects; part discovery is more complex than object discovery, primarily due to the ambiguity in the object part segmentation without applying additional constraints. Fortunately, for man-made objects, affordances are attached to objects at the part level. This observation implies the possibility of combining part discovery and affordance learning with minimal supervision. In this work, we integrate part discovery with affordance estimation, hoping that affordance information would help discover object parts sharing similar affordances.

**Unsupervised Geometric Primitive Modeling**   Whereas supervised geometric primitive abstraction methods (Mo et al., 2019a; Yang et al., 2020) require dense hierarchical annotations, unsupervised frameworks using cuboid-based (Tulsiani et al., 2017; Sun et al., 2019), superquadrics-based (Paschalidou et al., 2019; 2020), or other genus-zero-shape (Deng et al., 2020; Paschalidou et al., 2021) primitives discover structural information naturally embedded in the geometry. Recently, Yang & Chen (2021) unsupervisedly learn the cuboid-based shape abstraction with shape co-segmentation. Yet, it relies heavily on the ground-truth point normals for accurate abstraction and lacks semantic representation for object understanding. In our affordance discovery framework, we leverage the cuboidal regularization to refine the reconstructed affordance part, which distinguishes densely connected 3D parts by providing geometric prior, thus improving the affordance part discovery.

## 3 TASK DEFINITION

We formulate the new task *PartAfford* as discovering the part-level object affordance with the affordance set supervision. We define $K = 24$ common affordance categories $\mathcal{S} = \{s_k\}_{k=1}^{K}$, such as "sittable" and "openable," for object understanding. Input is given as a collection of $N$ objects

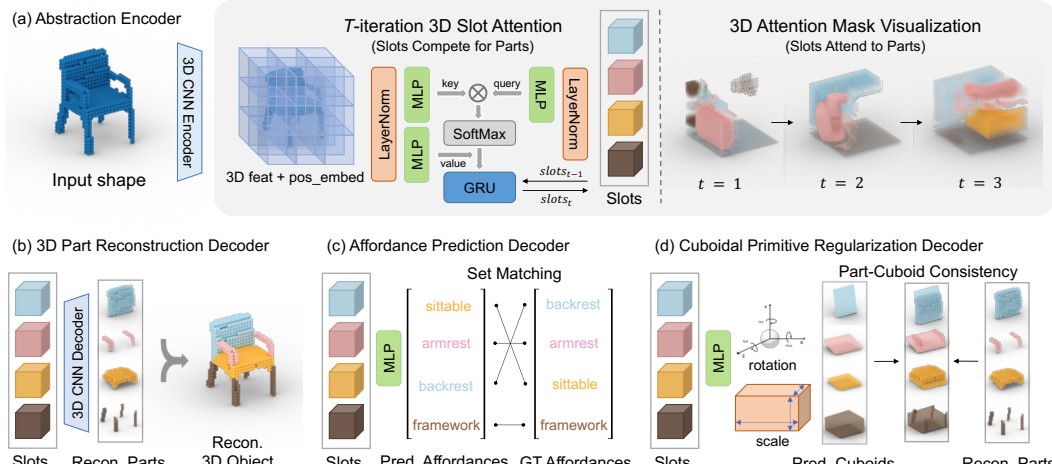

Figure 2: **Illustration of the proposed method for *PartAfford*.** Our model contains two main components: abstraction encoder and affordance decoder. (a) **Abstraction encoder** takes a 3D object as input, extracts features with 3D convolutional neural networks, and abstracts it into several slots. **Affordance decoder** with three branches jointly (b) reconstructs the 3D parts, (c) predicts affordance labels, and (d) regularizes cuboidal primitives. Each predicted cuboid in (d) wraps around the corresponding predicted object part in (b) tightly.

$\{o_i\}_{i=1}^N$ and their corresponding affordance set labels $\{\mathcal{A}_i\}_{i=1}^N$, where $\mathcal{A}_i = \{a_i^j\}_{j=1}^{J_i}$, $a_i^j \in \mathcal{S}$, and $J_i$ represents the number of distinct assigned affordances for each object $i$. *PartAfford* requires an algorithm to decompose each object into parts and discover the affordance corresponding to each object part. Fig. 1 illustrates the *PartAfford* task.

## 4 METHOD

We propose a novel framework for affordance discovery from 3D objects. It integrates unsupervised part discovery with affordance set prediction and geometric primitive abstraction; see Fig. 2. Given a 3D shape represented by voxel grids $\mathcal{V}$ of resolution $32^3$, our method first encodes the 3D shape into visual features and abstracts it into $M$ slots; each slot represents an abstracted high-level feature for downstream tasks. Next, we utilize a decoder with three branches to jointly (i) decode the features into parts, (ii) predict the affordance label, and (iii) regularize the parts with cuboidal primitives. Below, we describe in detail how each module is constructed and the loss design.

### 4.1 ABSTRACTION ENCODER

The encoder takes a 3D shape as input and abstracts part-centric latent codes in an unsupervised manner. It consists of a feature extraction module and a 3D slot attention module; see Fig. 2a.

**Feature Extraction** The feature embedding backbone encodes the input voxels and generates a $D = 64$ dimensional feature for each voxel. Following (Mescheder et al., 2019), voxels are encoded by five layers of 3D convolutional neural networks. The embedded feature is then augmented with absolute positional embedding (Locatello et al., 2020).

**3D Slot Attention** The 3D slot attention architecture, adapted from Locatello et al. (2020), serves as the part-centric representational bottleneck between the 3D feature embedding network and the downstream decoders. The encoded feature of a 3D shape is fed into an iterative attention module, where $M$ randomly initialized slots are updated for $T = 3$ iterations through a GRU (Cho et al., 2014). During each iteration, the attention coefficients are calculated by applying softmax normalization over the slots on the dot-product similarity between queries (*i.e.*, linearly-mapped 3D slot features) and keys (*i.e.*, linearly-mapped input features). The attention coefficients are then applied as weights for aggregating the values (*i.e.*, linearly-mapped input feature) and updating the slots.

Since the inputs-to-slots attention assignment is normalized over the 3D slots, those slots compete to attend to a clustering of the input 3D shape. Such clusterings are similar to human-defined parts on common objects. 3D slot attention masks naturally segment the object through iterations. An example of the learned 3D attention masks is shown in Fig. 2a.

## 4.2 Affordance Decoder

Shown in Fig. 2b-d, the affordance decoder takes part-centric slot features as input, followed by three branches for 3D part reconstruction, affordance prediction, and primitive regularization. The decoder parameter is shared across slots.

**3D Part Reconstruction** We design a 3-layer 3D transposed convolutional decoder followed by a single MLP layer to reconstruct voxel values $\hat{\mathcal{V}}^m$ and a voxel mask for each slot. The mask is normalized across slots with softmax, which generates a normalized mask $\hat{\Lambda}^m \in \mathbb{R}^{32 \times 32 \times 32}$. It is then used to compute the weighted sum of voxel values across slots and combine the reconstructed parts $\{\hat{\mathcal{V}}^m\}_{m=1}^M$ into a full 3D shape $\hat{\mathcal{V}}$: $\hat{\mathcal{V}} = \sum_{m=1}^M \hat{\Lambda}^m \hat{\mathcal{V}}^m$. The 3D part reconstruction branch is self-supervised by the reconstruction loss between original voxels $\mathcal{V}$ and reconstructed voxels $\hat{\mathcal{V}}$; we use the binary cross-entropy (BCE): $\mathcal{L}_{\text{recon}} = \text{BCE}(\mathcal{V}, \hat{\mathcal{V}})$.

**Affordance Prediction** We predict a one-hot affordance label for each slot with a two-layer MLP with sharing weights across slots for classification.

The affordance prediction branch is weakly-supervised as we do not provide affordance labels for each voxel. Instead, only the affordance label set for the entire object is used as the supervision signal. The model is tasked to learn the alignment between the abstracted parts and the affordance labels from set supervision.

As defined in Sec. 3, the ground-truth set of affordance labels for an input 3D object is denoted as $\mathcal{A}$. We denote $\hat{\mathcal{A}}$ as the set of slot affordance predictions. $\hat{\mathcal{A}}_\sigma$ is a permutation of elements in $\hat{\mathcal{A}}$, where $\sigma \in \mathfrak{G}$ and $\mathfrak{G}$ represent all $M!$ possible permutations. $\mathcal{L}_{\text{match}}$ is the pairwise matching cost between two sets, which can be calculated by mean square error (MSE) or cross-entropy:

$$\mathcal{L}_{\text{pred}} = \min_{\sigma \in \mathfrak{G}} \mathcal{L}_{\text{match}}(\mathcal{A}, \hat{\mathcal{A}}_\sigma). \tag{1}$$

Due to the order-invariant nature of slot modules, we apply the Hungarian matching algorithm (Kuhn, 1955), with Huber loss as the pair-wise matching cost, to calculate the set-based (Carion et al., 2020) affordance prediction loss in Eq. (1).

**Cuboidal Primitive Regularization** As a generalized soft k-means algorithm, the slot attention mechanism heavily relies on visual cues, such as the clustering of pixel colors on the image. As such, it cannot perform precisely in a crowded scene with overlapping objects even on a toy image dataset (Locatello et al., 2020). In the 3D voxel regime, segmenting objects into parts is challenging since every voxel is connected to neighboring voxels without distinguishable visual appearances.

Therefore, we introduce the cuboidal primitive regularization module, providing a geometric prior for segmentation: Human-made objects usually have geometric regularity, and cuboid is a concise structural representation for abstraction.

From each slot embedding, the cuboid abstraction module predicts a cuboid parametrized by two vectors (Yang & Chen, 2021): a scale vector $\boldsymbol{s} \in \mathbb{R}^3$ and a quaternion vector $\boldsymbol{r} \in \mathbb{R}^4$ for 3D rotation. Of note, we calculate the cuboid center from the weighted mean of voxel positions in the slot.

To evaluate how the predicted cuboid fits the reconstructed part in the $m$-th slot, we first compute the Euclidean distance $d_i^m$ from each voxel $\boldsymbol{p}_i^m$ to its closest cuboid face. Next, we calculate the weighted sum distance for all voxels, where the weight $v_i^m \in \hat{\mathcal{V}}^m$ is the reconstructed voxel value within $[0, 1]$. Additionally, we designed a binary surface mask $f(i)$ that masks out internal voxels in the loss. We also regularize the cuboid loss by adding a cuboid scale penalty term, *i.e.*, $L_1$ norm for the scale vector. Thus, the loss encourages a cuboid to wrap around a solid object part tightly. The regularization loss for all the slots is defined as:

$$\mathcal{L}_{\text{cuboid}} = \sum_m \left[ \lambda_{\text{scale}} \|\boldsymbol{s}^m\|_1 + \sum_i f(i) v_i^m d_i^m \right]. \tag{2}$$

**Total Loss** Taking together, the total training loss is the sum of 3D reconstruction loss, affordance prediction loss, and primitive regularization loss:

$$\mathcal{L}_{\text{total}} = \lambda_{\text{recon}} \mathcal{L}_{\text{recon}} + \lambda_{\text{pred}} \mathcal{L}_{\text{pred}} + \lambda_{\text{cuboid}} \mathcal{L}_{\text{cuboid}}, \tag{3}$$

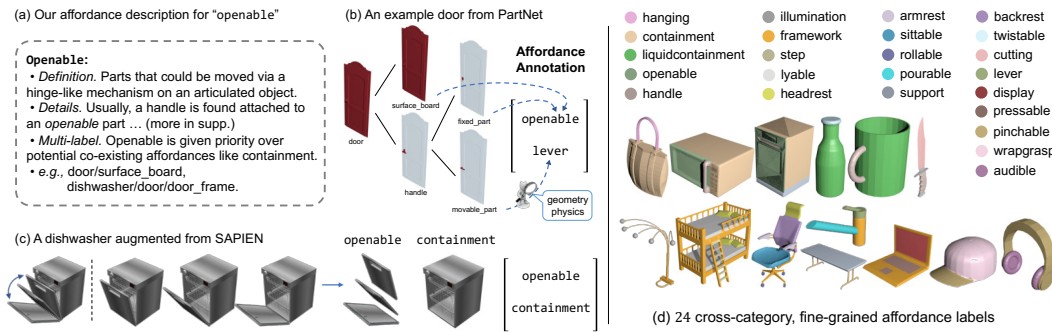

Figure 3: **Part affordance dataset.** (a) Description for the "openable" affordance to construct the mapping. (b) Given the part hierarchy of a door from PartNet (Mo et al., 2019b), we annotate its affordance labels by manual mapping and inspection. (c) Given a dishwasher from PartNet-Mobility (SAPIEN) (Xiang et al., 2020) and its kinematics, we rotate the door frame to include 3D objects with different articulation states. (d) Some exemplar 3D object models with color-coded affordance visualization.

where $\lambda_{\text{recon}}$, $\lambda_{\text{cuboid}}$, and $\lambda_{\text{pred}}$ are balancing coefficients.

Of note, with the current architecture design, for the first time, we demonstrate the capability of part-level affordance discovery from set labels. The exploration of more complex and practical modules is left for future work.

# 5 PART AFFORDANCE DATASET

To benchmark *PartAfford* and facilitate the research in affordance understanding, we construct a part-level 3D object affordance dataset. We focus on 24 cross-category, fine-grained affordance labels as shown in Fig. 3. The dataset is annotated with over 25,000 3D CAD models from the PartNet dataset (Mo et al., 2019b) and 625 articulated objects among 9 categories from the PartNet-Mobility dataset in SAPIEN (Xiang et al., 2020). Below, we describe how to define part affordances and generate affordance annotation. See *appendix* for more details.

Part affordances in our dataset are defined in terms of adjectives (*e.g.*, *sittable*) or nouns (*e.g.*, *armrest*), which describe how object parts could afford human daily actions and activities. We adopt certain common affordance categories from a comprehensive survey of visual affordance (Hassanin et al., 2021), *e.g.*, *containment*, *sittable*, *support*, *openable*, *rollable*, *display*, and *wrapgrasp*. However, they are coarse-grained–either at the object level or scene level. For example, "*openable*" only indicates whether an object can be opened, not on which object part can *afford* the object to be opened.

To pursue a fine-grained understanding of object affordance, we manually construct a one-to-multiple mapping from 479 kinds of object part labels defined at the finest granularity in Mo et al. (2019b) to 24 potential affordance labels, given the detailed affordance description. We provide expert-defined descriptions for 24 affordances to guarantee the quality and consistency of the mapping construction. An example is shown in Fig. 3a.

Given the part hierarchy of a 3D object, we can get the corresponding affordance annotation by mapping. We also perform a manual inspection to correct the affordance labels, especially for some fine-grained parts, according to their specific geometry and physics properties. For instance, different door handles will be mapped to different affordance labels (twistable, lever, *etc*.) according to how they should be operated (Fig. 3b).

The PartNet dataset does not contain articulation information, making affordances such as *openable* not geometrically distinguishable. Therefore, we generate a set of shapes with *openable* affordance from the PartNet-Mobility dataset by capturing 3D shapes with various opening angles (Fig. 3c).

As can be seen from Fig. 3d, each affordance type–due to its cross-category nature–may be found on various object part instances. For example, *openable* is usually afforded by rotatable doors for unobstructed access. Under such criteria, the door frame of a dishwasher and the surface board of a door are both mapped to *openable*. Please refer to the *appendix* for a full list of all affordance categories, descriptions, and mapping examples.

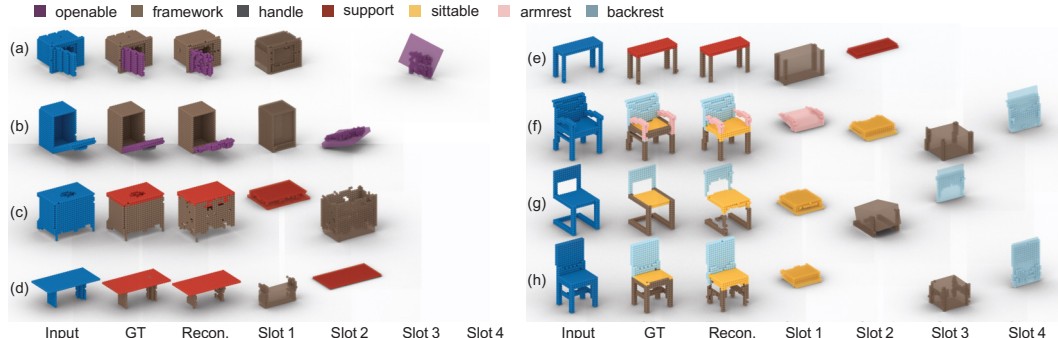

Figure 4: Qualitative results on three curated subsets: "openable" (a-b), "support" (c-e), and "sittable" (f-h).

# 6 EXPERIMENTS

In this section, we design and conduct comprehensive experiments to evaluate the proposed method. Fig. 4 visualizes our main results. We present both quantitative and qualitative comparisons of baseline models and other variants. We also evaluate the model generalization on novel shapes, analyze failure cases, and propose potential improvement directions. Please refer to the *appendix* for additional experimental settings, results, and analyses.

## 6.1 EXPERIMENTAL SETTINGS

**Benchmarks** To benchmark *PartAfford*, we curate different subsets of samples from our constructed dataset. Specifically, we study the subsets related to the most representative affordance categories "sittable," "support," and "openable" separately, where the subsets are created by collecting all cross-category objects that have the corresponding affordance label in our dataset. They contain 7 kinds of affordances from 8 object categories, covering $17,842/25K \approx 71\%$ instances. For each subset, we learn to distinguish all the affordance labels appear in the 3D objects. Note that although a part can have multiple affordances as mentioned in Sec. 5, we only keep the most prioritized affordance for each part to ease the ambiguities in learning.

**Evaluation Metrics** In *PartAfford*, we evaluate the performances of part discovery (clustering similarity), 3D reconstruction, and affordance prediction.

- Part Discovery: We use the Intersection over Union (IoU) to evaluate the part similarity. Specifically, we employ Hungarian matching to find the best matches between the reconstructed parts and ground truth parts using voxel IoU as the matching score. Then we compute the mean IoU (mIoU) by averaging the IoU between best matches.
- 3D Reconstruction: We evaluate the shape reconstruction quality by Mean Squared Error (MSE).
- Affordance Prediction: Following Locatello et al. (2020), we use Average Precision (AP) to evaluate the prediction accuracy. A correct prediction means an exact match of the affordance label set.

**Baselines and Ablations** Since we are the first to propose and formulate *PartAfford*, we have no previous work to make direct comparisons. Therefore, we compare our method with two designed baseline models and three variants:

- Slot MLP: a simple MLP-based baseline where we replace Slot Attention with an MLP that maps from the learned feature maps (resized and flattened) to the (now ordered) slot representation.
- IODINE: a baseline where we replace the Slot Attention with an object-centric learning method IODINE (Greff et al., 2019) to abstract and cluster the encoded feature.
- Ours w/o Afford & Cuboid: our model variant that only keeps the 3D part reconstruction branch.
- Ours w/o Afford: our model variant without the affordance prediction branch.
- Ours w/o Cuboid: our model variant that discards the cuboidal primitive regularization branch.
- Ours Full: our full model with all branches.

## 6.2 IMPLEMENTATION DETAILS

**Learning Strategy** To stabilize the training, we split the training into two stages. In the first stage, we train the decoder only with 3D part reconstruction and affordance prediction branches. In the

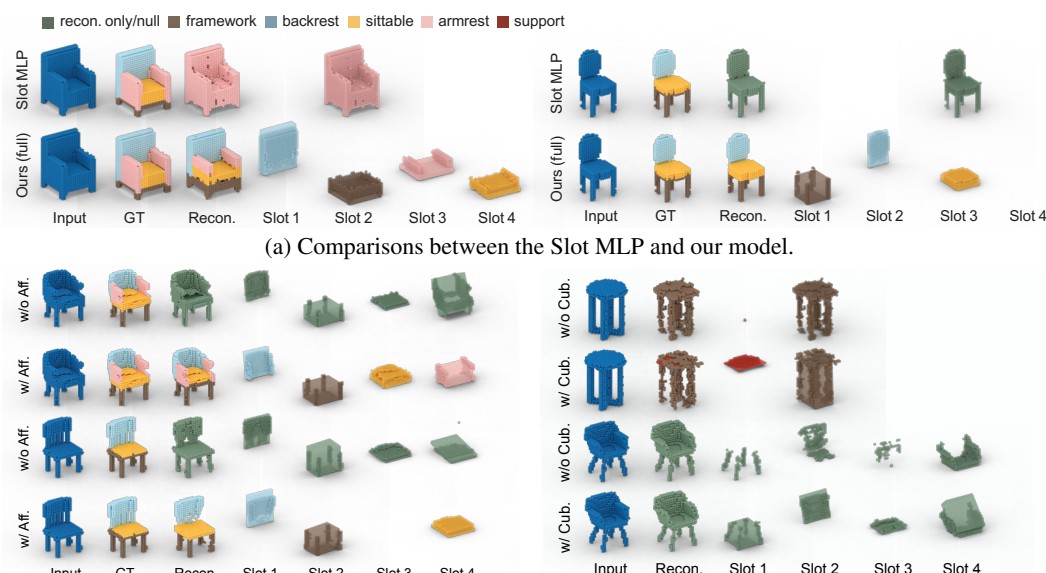

(a) Comparisons between the Slot MLP and our model.

(b) Comparisons between the models without and with affordance prediction branch. The affordance prediction facilitates the abstraction of 3D shape (rows 1-2) and elimination of spare slots (rows 3-4).

(c) Comparisons between the models without and with cuboidal primitive regularization branch. The cuboidal primitive regularization helps to guide and improve the segmentation.

Figure 5: Qualitative comparison results.

second stage, the cuboidal primitive regularization branch is incorporated into joint training with a lower learning rate.

**Hyperparameter** We set learning rate as $4 \times 10^{-4}$ for the first stage, $2 \times 10^{-4}$ for the second stage, and apply Adam optimizer (Kingma & Ba, 2014) for optimization. It takes $5 + 9$ hours on $4$ RTX A6000 GPUs for two-stage full-model training of "sittable"-related objects. For slot attention, we empirically set the number of GRU iterations $T = 3$. We set the number of slots to the maximal number of affordance labels that appear in each training subset. For example, we learn the "sittable" with $4$ slots, "support" with $2$ slots, and "openable" with $3$ slots. Appendix C.1 further discusses choices of the number of slots. For the joint loss weight, we set $\lambda_{\mathrm{recon}} = 1.0$, $\lambda_{\mathrm{pred}} = 0.5$, $\lambda_{\mathrm{cuboid}} = 0.1$.

## 6.3 Results and Analysis

Tabs. 1 and 2 tabulate the quantitative performances of all the models under different settings. Fig. 4 qualitatively shows the capability of our method and Fig. 5 compares different models. Below, we summarize some key findings:

1. The proposed method achieves the best overall performance on the *PartAfford* task, especially in the part discovery (mean IoU) where it outperforms the baselines by a large margin. This demonstrates the outstanding abstraction capability of our approach given the weak supervision. From Fig. 4, we can see our method can discover the detailed part-level representation with their aligned affordances for the 3D objects.
2. The most challenging part of *PartAfford* lies in the part discovery; it is also where our model differentiates from other baselines. For example, Tab. 1 shows that Slot MLP achieves the best affordance prediction performance (AP) but fails in the part discovery (mean IoU) and 3D reconstruction (MSE). As also shown in Fig. 5a, the Slot MLP cannot segment the object input to parts due to the lack of abstraction capability.
3. Affordance prediction branch significantly escalates the part discovery performance since it helps to learn part-affordance correspondence from the affordance composition, which provides contrasts to distinguish different parts among the training objects. Our qualitative results also show that the affordance prediction facilitates the abstraction of 3D shape (*e.g.*, rows 1-2 of Fig. 5b) and elimination of spare slots (*e.g.*, rows 3-4 of Fig. 5b);
4. Cuboidal primitive regularization branch also boosts the part discovery, especially when affordance prediction is unavailable. This demonstrates that geometric priors play a crucial role in

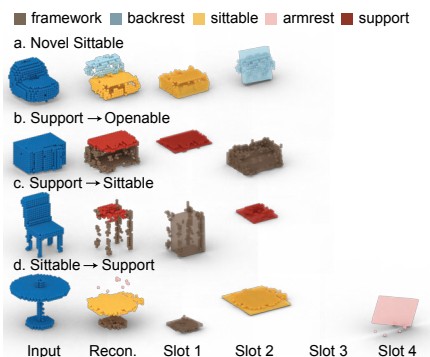

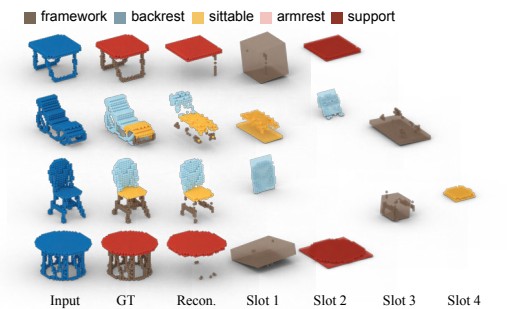

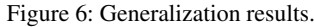

Figure 6: Generalization results.

Figure 7: Failure cases.

segmentation when data are not diverse enough. From Fig. 5c, we can see the cuboidal primitive regularization helps to segment better primitives and avoid scattered voxels.

### 6.4 MODEL GENERALIZATION

With the cross-category nature of affordance, we qualitatively test how the learned model can be generalized to novel objects and unseen categories. We conduct model generalization experiments by testing hard examples or objects from other categories. Examples from Fig. 6 demonstrate the learned model could be generalized to

Table 1: **Quantitative results on "sittable."** We evaluate the mean IoU (mIoU), mean squared error (MSE), and average precision (AP) on included objects.

| Model | mIoU (%) ↑ | MSE ↓ | AP (%) ↑ |
|---|---|---|---|
| Slot MLP | 21.5 | 0.0150 | **94.5** |
| IODINE | 49.2 | 0.0102 | 92.5 |
| Ours w/o Afford & Cuboid | 31.5 | 0.0112 | N/A |
| Ours w/o Afford | 39.4 | 0.0100 | N/A |
| Ours w/o Cuboid | 55.3 | 0.0102 | 92.7 |
| Ours (full) | **57.3** | **0.0097** | 92.9 |

objects with diverse shapes. We show the results of testing the learned model on a novel object shape (bean bag) (a) from Fu et al. (2021) and unseen categories (b-d). For example, (b) shows the result of learning with "support" and testing on an "openable" object (*i.e.*, a microwave). Despite the imperfect reconstructions, partly due to the reconstruction bottleneck's impact on disentanglement quality (Engelcke et al., 2020), the learned model can identify the functional parts given novel objects.

### 6.5 FAILURE CASES

We show some failure cases of our method in Fig. 7. For "sittable" and "support," the failures are commonly caused by (i) the difficulties in reconstructing the fine-grained details of 3D objects with novel shapes; (ii) certain parts that violate the cuboid assumption, and thus hurt other components.

For objects in "openable" category, our model cannot discover and reconstruct "handle," as shown in Fig. 4. This is because the objects with related affordances come from various object categories with diverse shapes, making it challenging for the model to capture such complex mixtures of distributions and reconstruct fine-grained 3D shapes, especially tiny parts. This points out future directions to better understand object parts (*e.g.*, segment, reconstruct), and po-

Table 2: **Quantitative results on "support" and "openable."**

| | Model | mIoU (%) ↑ | MSE ↓ | AP (%) ↑ |
|---|---|---|---|---|
| support | Slot MLP | 36.8 | 0.0099 | 91.6 |
| | Ours w/o Afford | 34.8 | 0.0092 | N/A |
| | Ours w/o Cuboid | 51.3 | 0.0087 | **95.2** |
| | Ours (full) | **52.7** | **0.0085** | 95.1 |
| openable | Slot MLP | 21.0 | 0.0130 | **70.8** |
| | Ours w/o Afford | 19.9 | 0.0104 | N/A |
| | Ours w/o Cuboid | 46.7 | 0.0097 | 55.8 |
| | Ours (full) | **47.6** | **0.0093** | 60.4 |

tentially an interactive learning framework to learn beyond geometry and appearance.

## 7 CONCLUSION

We present *PartAfford*, a new task in visual affordance research that aims at discovering part-level affordances from 3D shapes. We propose a novel learning framework that discovers part-level affordances by leveraging only the affordance set supervision and geometric primitive regularization. With comprehensive experiments and analyses, we point out potential directions for incorporating visual appearance to facilitate better shape abstraction and combining it with an active learning approach for efficient affordance learning.

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

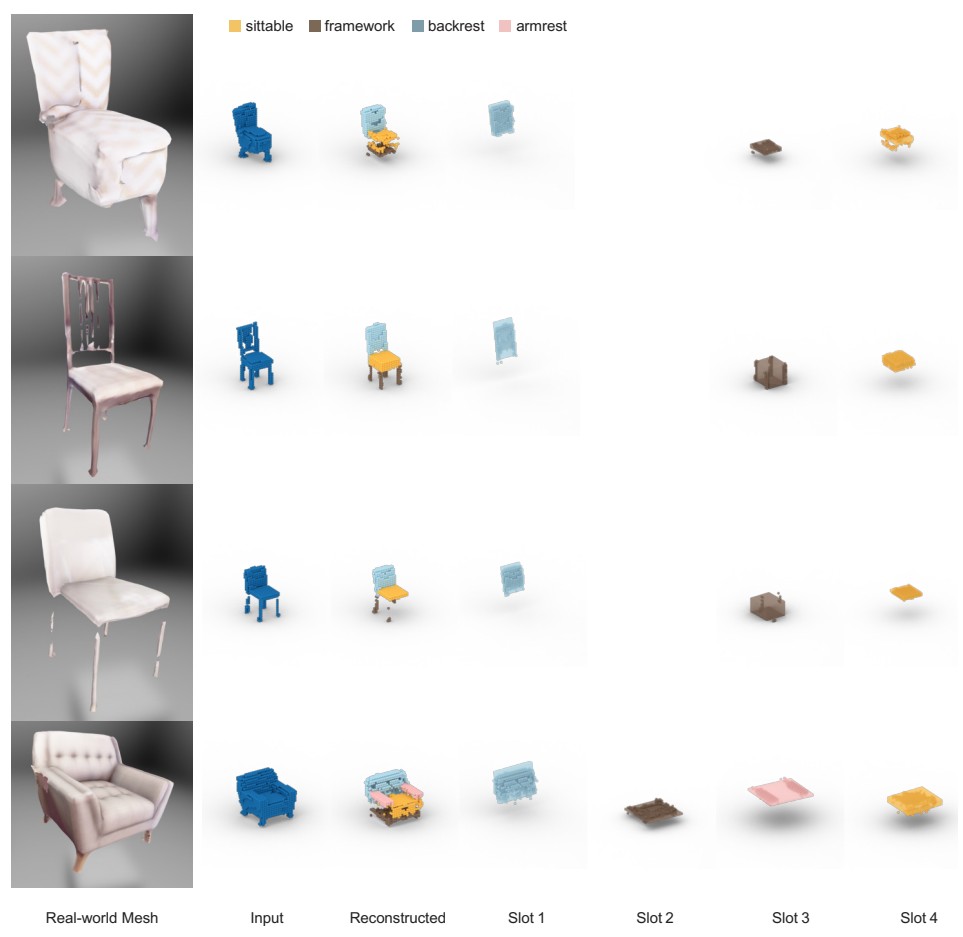

Figure 8: Qualitative results on real-world objects (chairs and sofas) from Replica (Straub et al., 2019).

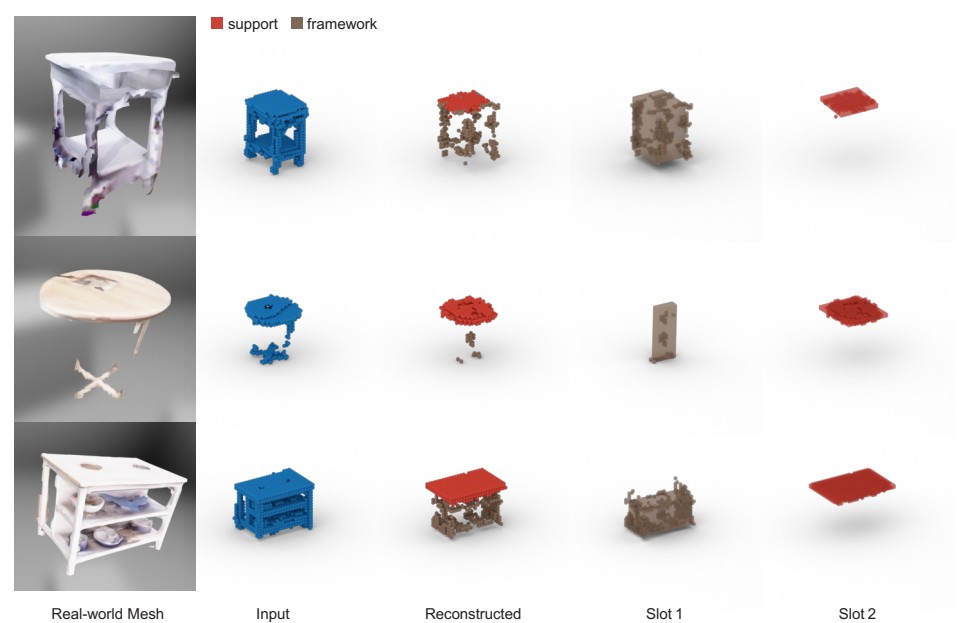

Figure 9: Qualitative results on real-world objects (tables and cabinets) from Replica (Straub et al., 2019).

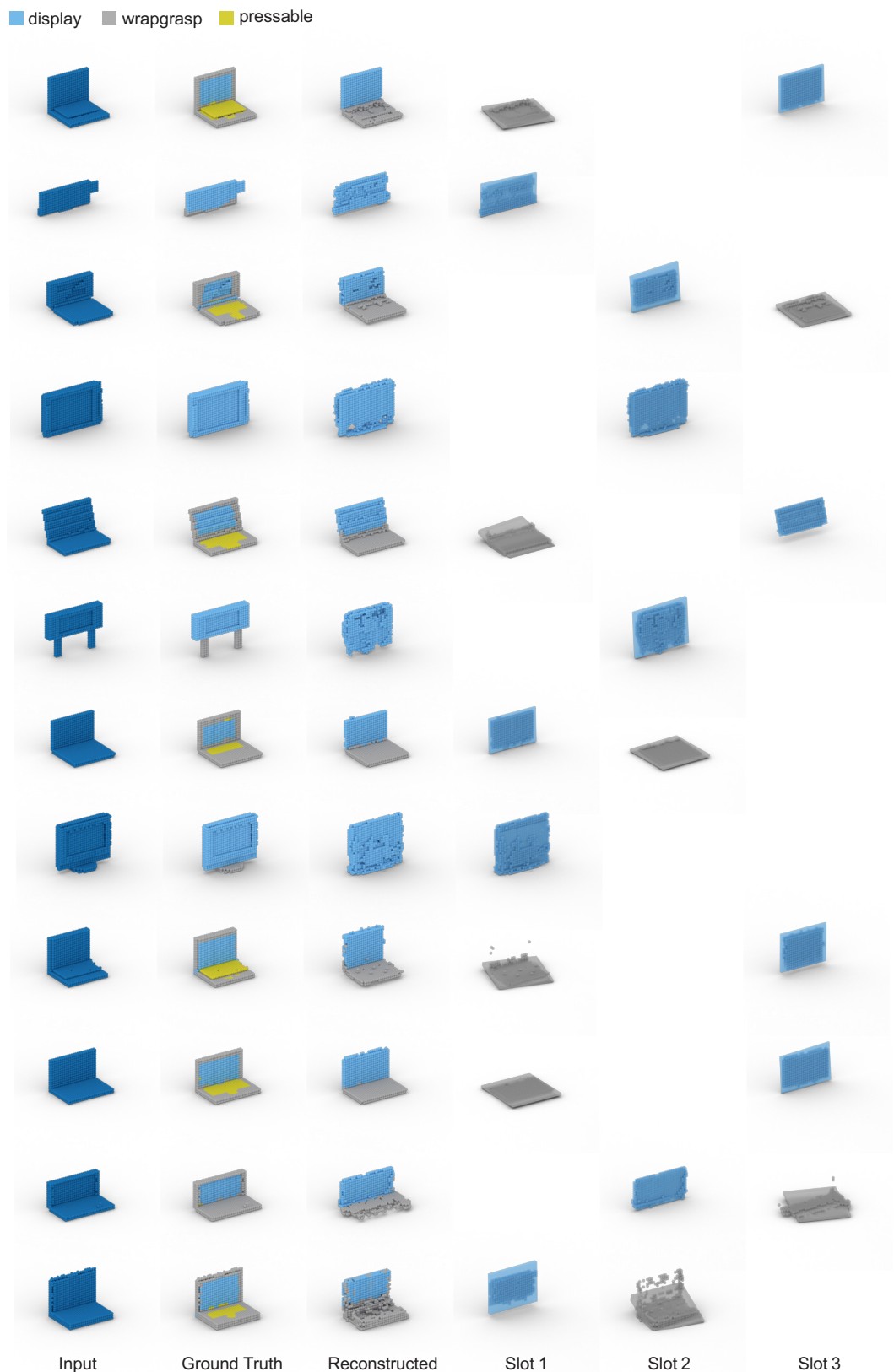

Figure 10: Qualitative results on objects (laptops and displays) in our "display" subset.

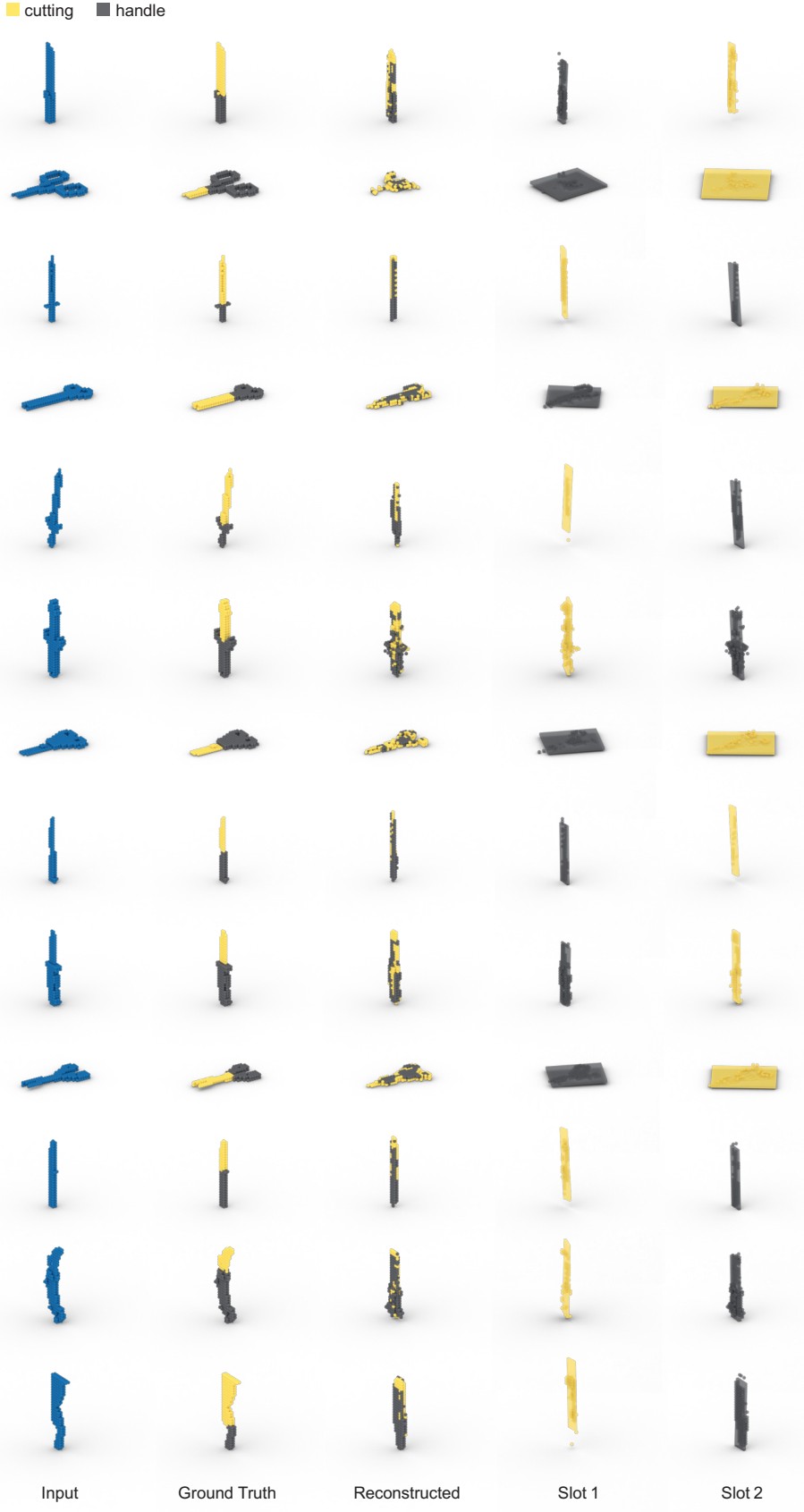

Figure 11: Qualitative results on objects (knives and scissors) in our "cutting" subset.

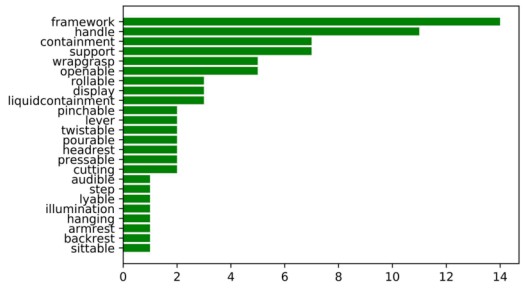 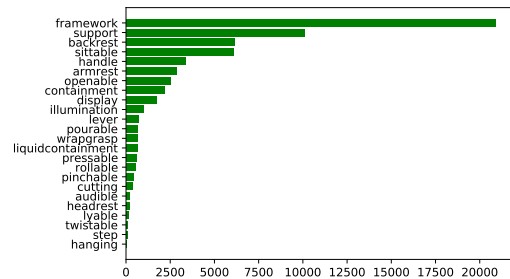

(a) Number of object categories covered by each affordance category.

(b) Number of object instances covered by each affordance category.

Figure 12: Statistics of our part affordance dataset.

# A EXPERIMENTS ON MORE AFFORDANCES

We conduct further experiments and report performance on more affordance categories on the long tail of the dataset distribution. More specifically, we include the subsets "display" and "cutting". The "display" set includes part-level affordances {*display, wrapgrasp, pressable*} from 1,441 object instances, most of which are laptops and displays. The "cutting" set includes affordances {*cutting, handle*} from 641 object

Table 3: **Quantitative results on the "display" subset and the "cutting" subset.**

|  | Model | mIoU (%) ↑ | MSE ↓ | AP (%) ↑ |
|---|---|---|---|---|
| display | Slot MLP | 17.2 | 0.0163 | **97.2** |
|  | Ours w/o Afford & Cuboid | 14.8 | 0.0146 | N/A |
|  | Ours w/o Cuboid | 47.0 | 0.0140 | 80.6 |
|  | Ours (full) | **47.6** | **0.0135** | 82.0 |
| cutting | Slot MLP | 21.6 | 0.0030 | 97.7 |
|  | Ours w/o Afford & Cuboid | 27.8 | 0.0028 | N/A |
|  | Ours w/o Cuboid | 28.1 | 0.0027 | 96.9 |
|  | Ours (full) | **29.8** | **0.0026** | **97.7** |

instances, most of which are knives and scissors on the long tail of the dataset distribution. Quantitative results are shown in Tab. 3 and qualitative results are shown in Fig. 10 and Fig. 11. Our method still shows a significant advantage over the baseline methods on two core metrics: mIoU for part affordance discovery and MSE for 3D reconstruction.

# B ATTENTION VISUALIZATION

Following the settings in Locatello et al. (2020), we train the models using $T = 3$ attention iterations in the slot attention module. In Fig. 13, we observe that each slot gradually attends to the correct part of the object as the number of attention iterations increases. The attention mask at $t = 3$ depicts the silhouette of the reconstructed shape and parts.

# C FURTHER ANALYSIS & DISCUSSION

## C.1 NUMBER OF SLOTS

We set the number of slots as the maximal number of affordance labels that appear in one subset, which is different from previous object-centric learning algorithm (Locatello et al., 2020), where the number of slots could be arbitrary. This is because when we increase the number of slots, we also increase the ambiguities of the affordance composition and set matching at the same time. It prevents the model from learning accurate correspondence between affordance labels and parts. As shown in Fig. 14, the model learns the "sittable" in a "null" slot.

## C.2 SELECTION OF STUDIED AFFORDANCES

Although we annotate objects with 24 affordance categories, we benchmark *PartAfford* only on three subsets with 7 kinds of affordances in this work. The main reason is that we find it much more challenging to discover part-level affordance for some other affordance categories. For example, "rollable" and "cutting" usually connect to tiny object parts that are challenging to be segmented from objects, "illumination" and "display" cannot be distinguished from the objects without a deeper understanding of the visual appearance and reflections, "pressable" and "pinchable" require richer interactions to be discovered. In summary, it is either especially challenging to segment or requires more than geometric information (*e.g.*, active interactions) to discover the other affordance categories.

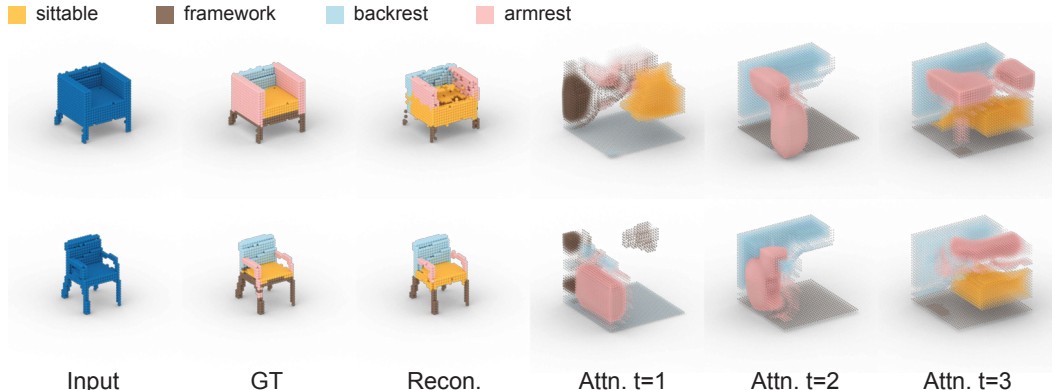

Figure 13: Example affordance discovery model trained with $T = 3$ attention iterations. Attention is visualized in various colors and point radius. The point is colored according to the predicted affordance label of the slot which attends the most to the point. Point radius positively depends on the maximal attention value across the slots. We use trilinear upsampling to rescale the attention mask to the input resolution ($32 \times 32 \times 32$).

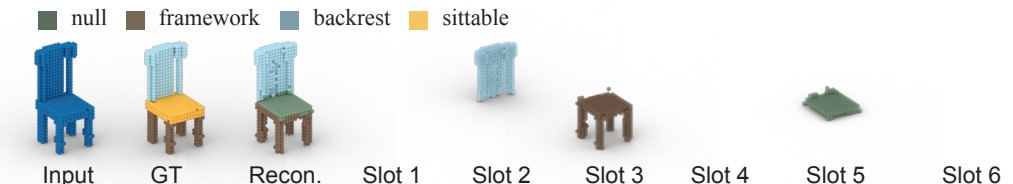

Figure 14: Model performance when the number of slots increases. The model learns the "sittable" part in a "null" slot.

We believe that future solutions to our task could involve those interaction cues to take the best of both worlds, further improving the performance and extending the scope. Our solution could also provide an intuitive prior for active learning methods, which could boost learning efficiency compared to learning from scratch. We hope to further explore the learning of these affordances in the future.

### C.3 Ambiguities in Affordance Learning

Affordance is naturally ambiguous since the functions of object parts are rich. This work provides a well-defined benchmark to study how to learn affordance from accurate affordance definition and sparse set supervision. In another work, (Deng et al., 2021) models the multiple affordances with a mixture of distributions. Our dataset has supported learning multiple affordances per part; we annotate a single part with multiple affordances (see details in Sec. 5). Yet, as the first step towards affordance discovery, our current method focuses on learning the most important affordance. We believe our current learning framework could also be extended to learn multiple affordances by switching the one-hot affordance label to multi-hot labels. We also suggest further fine-tuning the multi-hot affordance prediction based on the one-hot version, since the affordance prediction of single-hot labels appears significant to guide the slot competition from our ablation study.

### C.4 Potential Impacts for Vision & Robotics Community

By learning to discover the part-level affordance, our model could facilitate the understanding of human-object interaction and object manipulation. As shown in Fig. 15, the learned affordance could be applied to synthesize potential human actions and interactions with various 3D objects. Additionally, our work could provide priors for robotic cross-category object manipulation and task planning (Toussaint et al., 2019; Garrett et al., 2020; 2021). For example, Hadjivelichkov et al. (2022) demonstrates a single reference image of an object with annotated affordance regions can help the robot to use the affordance skill in the real-world setting, where our method can provide generalizable affordance information on the 3D shapes without the need for manual annotation.

|        | sittable | support | frame. | contain. | liquid. | openable | display | cutting |
|--------|----------|---------|--------|----------|---------|----------|---------|---------|
| Orig.  | 72.44    | 82.92   | 74.74  | 54.01    | 20.34   | 58.62    | 87.26   | **83.29** |
| Scaled | **74.95** | **83.68** | **77.87** | **75.01** | **69.42** | **66.26** | **90.03** | 79.67   |

|        | backrest | armrest | press. | handle | illum. | wrapgrasp | lyable | headrest |
|--------|----------|---------|--------|--------|--------|-----------|--------|----------|
| Orig.  | 80.51    | 73.19   | 78.70  | 38.09  | 24.57  | 63.07     | **46.24** | **34.73** |
| Scaled | **81.36** | **73.63** | **83.57** | **49.31** | **24.59** | **71.48** | 45.32  | 32.76    |

|        | rollable | pourable | twist. | lever | pinch. | audible |
|--------|----------|----------|--------|-------|--------|---------|
| Orig.  | 53.33    | 68.15    | 40.24  | 60.34 | 62.07  | **56.67** |
| Scaled | **58.43** | **73.89** | **53.97** | **64.24** | **66.84** | 46.11   |

Table 4: Supervised affordance segmentation results (category mIoU %). "Orig." refers to the original PartNet dataset with 3D shapes scaled to a unit bounding sphere. "Scaled" refers to rescaling 3D shapes to real-world dimensions. "avg" refers to shape average Intersection-over-Union (IoU).

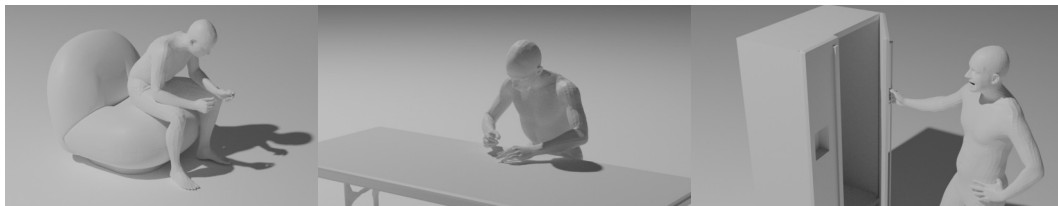

Figure 15: 3D Human synthesis conditioned on inferred part affordance using (Hassan et al., 2021).

## D    SUPERVISED AFFORDANCE ESTIMATION

Apart from the *PartAfford* task, we also benchmark the state-of-the-art 3D auto-encoder for the supervised affordance segmentation task in a cross-category fashion on our proposed dataset. Evaluation is reported on the Intersection-over-Union (IoU) metric following Mo et al. (2019b), as shown in Table 4.

Note that the baseline is trained and tested on all object and affordance categories. In contrast, semantic segmentation in PartNet (Chang et al., 2015; Mo et al., 2019b) is trained on each object category separately. Results show that the algorithm could still achieve high performance, which demonstrates that our affordance annotation is reasonable and consistent.

PartNet normalizes every object shape into a unit bounding sphere. This is not realistic since objects in different categories may have very different dimensions. For example, a mug is much smaller than a bed. If normalized in the same way, cross-category training performance may be hurt to some extent. Thus, we calculate the average real-world 3D dimensions of each object category from metadata of ShapeNet (Chang et al., 2015) and scale the point cloud in our part affordance dataset according to its object category.

## E    MODEL GENERALIZATION TO UNSEEN OBJECTS

To further demonstrate the generalizability of our methods qualitatively, we additionally provide examples for real scanned and more diverse objects from the Replica (Straub et al., 2019) dataset, results shown in Fig. 8 and Fig. 9. Although the scan quality is not perfect, our learned model generalizes well and can reconstruct and identify the functional parts given novel objects.

## F    DETAILS ON EXPERIMENTS

**Benchmarks**    Below, we describe the statistics of objects and the related affordances for these subsets; we discuss detailed reasons about why we choose these three subsets in Appendix C.2.

- "Sittable": We collect all object instances that have affordance "sittable"; most of them are chairs and sofas. Their part-level affordances belong to the set $\{sittable, backrest, armrest, framework\}$. We split the training, validation, and test set in the ratio of $7:1:2$. In total, we have $5,093$ instances for training and $1,457$ for test.

- "Support": We collect objects with affordance "support", mainly from categories table and cabinet. Their affordances belong to $\{support, framework\}$. There are $7,974$ instances for training and $2,279$ instances for test.
- "Openable": This subset contains objects from frige, dishwasher, washing machine, and microwave. Their affordances belong to $\{openable, framework, handle\}$. There are $807$ instances for training and $232$ instances for test.

**Data Augmentation** To enrich affordance compositions, we augment the training data by randomly removing certain object parts with corresponding affordance labels.

## G    MORE MODULE ABLATION

### G.1    SOFT K-MEANS

Slot attention, as a generalized soft k-means algorithm, could be reduced to the soft k-means algorithm according to Locatello et al. (2020). It turns out that the reduced model can achieve $23.0\%$ Mean IoU on "sittable" objects, which is slightly better than the Slot MLP baseline but significantly worse than our full model using slot attention ($57.3\%$). It reconstructs the whole shape with a quality (MSE: 0.0096) on par with the full model (MSE: 0.0097). Its affordance set prediction accuracy is the lowest among all models (AP: $0.87$). Overall, the soft k-means algorithm cannot effectively segment the shape or discover affordance parts.

## H    ADDITIONAL EXPERIMENTAL RESULTS

We show additional qualitative results for our full model's affordance discovery results on "sittable" (Fig. 16), "support" (Fig. 17), "openable" (Fig. 18) objects, respectively.

## I    PART AFFORDANCE DATASET

Here we report more statistics and details on how we define and annotate the affordance labels in our dataset. As shown in Fig. 12, most of the affordance categories are annotated on more than 2 kinds of objects. Our curated subsets cover the most common affordance categories (top 7), as well as some rare affordance categories (*e.g.*, "cutting") on the long tail.

### I.1    PRINCIPLES FOR AFFORDANCE ANNOTATION

To keep annotations consistent across object categories, we design a guideline for affordance annotation. Below are some general principles to annotate a leaf part of an object instance with affordances.

*Multiple affordances.* A part can afford multiple kinds of human actions. For example, the seat of chair could afford *sittable* for resting of human body or *support* if one wants to place some books on it. We refer to ConceptNet (Speer et al., 2017), a giant knowledge database, for annotating common usage of object parts.

*Prioritized fine-grained affordances.* When there are multiple affordances labels for a part, we give the more fine-grained affordance label higher priority. For the chair seat in the example above, *sittable* is prioritized compared with *support* as it is a fine-grained support affordance for body resting.

*Articulation-related affordances.* The PartNet dataset does not contain articulation information, which makes affordances such as *openable* not geometrically distinguishable. Thus, we also generate a set of shapes with *openable* affordance from the PartNet-Mobility dataset by capturing 3D shapes with various opening angles. More geometric variation helps models to learn articulation-related affordances.

### I.2    AFFORDANCE DESCRIPTIONS

Our description for each affordance contains a brief definition, some supplemental clarification and priority statements if needed, and some example leaf nodes in the part hierarchy of various reasonable objects (full path from root to leaf).

***sittable***:  Indicates whether the object can be used for sitting. Anything sittable of course affords support, and the requirement for a supporting object to be sittable is that it must be both comfortable and safe for human seating. For example, a table is not sittable despite affording support because it is not comfortable. *Sittable* is given priority over potential co-existing affordances like *support*.

*E.g.*, chair/chair_seat/seat_surface.

***support***:  A trait of objects which can safely keep other objects on top of themselves. Common characteristics of support-affording objects are that they are flat and can support multiple objects at once. A key distinction between support-affording objects and non-supporting objects is that support-affording objects will remain stable when other objects are placed on them. For example, a table is a supporting object because it is just as stable with objects on it as it is without. However, a stack of plates is not supporting because they become more unstable as you add more plates.

*E.g.*, table/regular_table/tabletop,
    storage_furniture/cabinet/shelf,
    bed/bed_unit/bed_sleep_area.

***openable***:  Parts which may be moved with a hinge-like mechanism on an articulated object. Openable objects do not need to afford handles, but usually handles can be found attached to the *openable* part. *Openable* parts are distinct from other moving parts in that they need to swing to some degree to be moved. *Openable* is given priority over potential co-existing affordances like *containment*.

*E.g.*, door/door_body/surface_board,
    dish_washer/body/door/door_frame,
    microwave/body/door.

***backrest***:  Objects which are reasonably designed for providing support to a person's back. By reasonably designed, we mean either specifically (as in the back of a chair) or can afford back support if a person wanted to sit upright (like a headboard).

*E.g.*, chair/chair_back/back_support.

***armrest***:  Objects which are specifically designed to support an arm. For example, a table can support a variety of things and is thus not an armrest. A chair's arm is the perfect size for a human arm, so it must be an armrest.

*E.g.*, chair/chair_arm.

***handle***:  An object extension which affords the ability to open an attached 'openable' part. Handles are mostly grabbed with hand-wrapping, so it's important to only afford 'handle' to parts which are specifically involved in an opening mechanism, like a door handle.

*E.g.*, storage_furniture/cabinet/cabinet_door/handle,
    table/regular_table/table_base/drawer_base/
    cabinet_door/handle,
    mug/handle,
    dish_washer/body/door/handle,
    bag/bag_handle.

***framework***:  Any object segment which either: a) helps to define the shape of the object as a whole or b) is an unaffording extension of the object or connector for other segments. For example, a hinge affords framework because it is integral to connects the door to the door frame. Overall, this affordance is the most general of all, so it should only be used when it clearly applies to either case of the definition. For example, most handles do not afford framework because it is both a small part of an object's shape and already affords *handle*. It has the least priority among potential co-existing affordances.

*E.g.*, chair/chair_base.

***containment***:  An affordance of object which can store physical items. The size of the physical items does not matter, so long as they are not too small. For example, anything that can only contain objects smaller than say a marble do not afford containment. Also, items that afford containment must afford security to the items they contain, such that they will not fall out.

*E.g.*, table/regular_table/table_base/drawer_base/drawer,
    storage_furniture/cabinet/drawer,
    mug/container,
    trash_can/container,
    refrigerator/body,
    bowl/container, bag/bag_body,
    bottle/normal_bottle/body.

***liquidcontainment***:   A more specific version of the containment affordance. Objects that afford liquid-containment must be able to safely contain liquid. Examples of these are bottles, bath tubs, *etc*.

*E.g.*, mug/body,
    bottle/normal_bottle/body.

***display***:   Something which visualizes information for a useful purpose. Examples of these would be monitor screens or a clock surface.

*E.g.*, display/display_screen/screen,
    laptop/screen_side/screen,
    clock/table_clock/clock_body/surface.

***cutting***:   The quality of being able to slice through other objects. Certain things that can cut are not considered to have *cutting* affordance if it was used against its intended purpose, like smashing a glass vase. Cutting is only afforded to objects which are specifically designed for cutting, like a blade-edge.

*E.g.*, cutting_instrument/knife/blade_side,
    scissors/blade_handle_set/blade.

***pressable***:   A mechanical feature of objects which either have buttons or can interact with a finger. Good examples of these are keyboard keys.

*E.g.*, keyboard/key.

***hanging***:   A part which can be hung on another object. These parts almost always only serve the purpose of hanging the rest of the entire object. An example of this would be a shoulder strap for a handbag.

*E.g.*, bag/shoulder_strap.

***wrapgrasp***:   The 'wrap-grasp' trait is afforded by parts which are explicitly meant to be grabbed in a hand-wrapping motion. Just because a hand can wrap around an object part does not mean it affords wrap-grasp. It must be useful to grip the part in this way. An example of this would be a ladder rung, which a person is meant to wrap their hand around to climb the ladder.

*E.g.*, cup,
    bed/ladder/rung.

***illumination***:   The affordance of light emission. This only applies to object parts which are meant to light up a broad area. For example, a monitor screen does not afford illumination despite emitting light because it is not supposed to be used to light up the area around it.

*E.g.*, lamp/table_or_floor_lamp/lamp_unit
    /lamp_head/light_bulb.

***lyable***:   Indicates that a human can comfortably rest his/her entire body on the object. These objects are usually flat with a soft surface. *Lyable* is given priority over potential co-existing affordances such as *sittable* and *support*.

*E.g.*, bed/bed_unit/bed_sleep_area/mattress.

***headrest***:   An extension of an object which is oriented so a human head can rest comfortably on it. Examples of these are chair headrests or bedframe headrests.

*E.g.*, bed/bed_unit/bed_sleep_area_pillow,
    bed/bed_unit/bed_frame/headboard.

***step***: A part which affords the human foot climbing or resting functionality. For example, a ladder rung is a step because it affords climbing with both hands and feet. A foot pedestal is also a step because it can be stood on or feet can be rested on it.

*E.g.*, bed/ladder/rung.

***pourable***: Meant for parts which liquid can flow out of. Things that are pourable may also be dependent on a mechanism for controlling flow, like a bottle cap or a knob.

*E.g.*, bottle/normal_bottle/mouth,
    bottle/jug/body.

***twistable***: These objects can either be detached or provide special functionality by twisting them in a clockwise or counterclockwise motion. Examples include bottle caps and knobs.

*E.g.*, bottle/normal_bottle/lid,
    bottle/normal_bottle/mouth.

***rollable***: A part which can roll to move around. Exceptions to this affordance are objects which roll but stay fixed in place, like a rocking chair.

*E.g.*, wheel.

***lever***: Any handle which can rotate up to a point. For example, knobs rotate but are not levers because they do not provide handles. Levers must be treated differently from twistable objects or handles because if they are twisted too much they will break.

*E.g.*, lever.

***pinchable***: An object which is small enough such that it can be manipulated by pinching with two or more fingers. Things that are pinchable must not be heavy, and they usually fit inside the palm of a hand.

*E.g.*, earbud.

***audible***: Anything which emits sound. This does not include sound emitted indirectly, such as a door creaking when opened, which makes sound as a side-effect.

*E.g.*, headphone/padding.

### I.3 LICENSE

Our dataset is annotated based on PartNet (v0) (Mo et al., 2019b) and PartNet-Mobility (v2.0) (Xiang et al., 2020), both of which are licensed under the terms of the MIT License.

## J CODE AND DATA

Code, data, and instructions needed to reproduce the main experimental results are available in the supplementary materials.

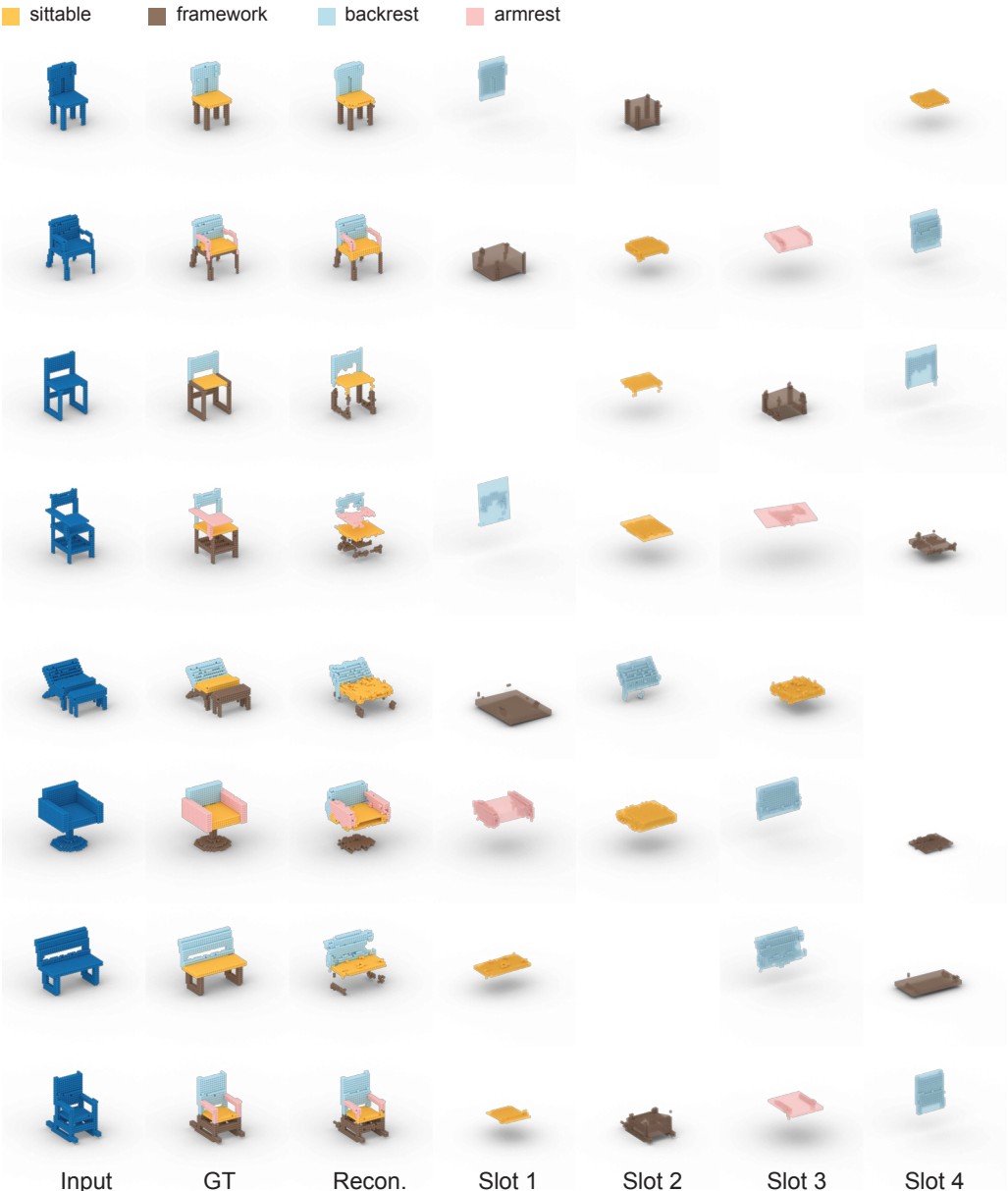

Figure 16: Additional qualitative results on objects in our "sittable" subset.

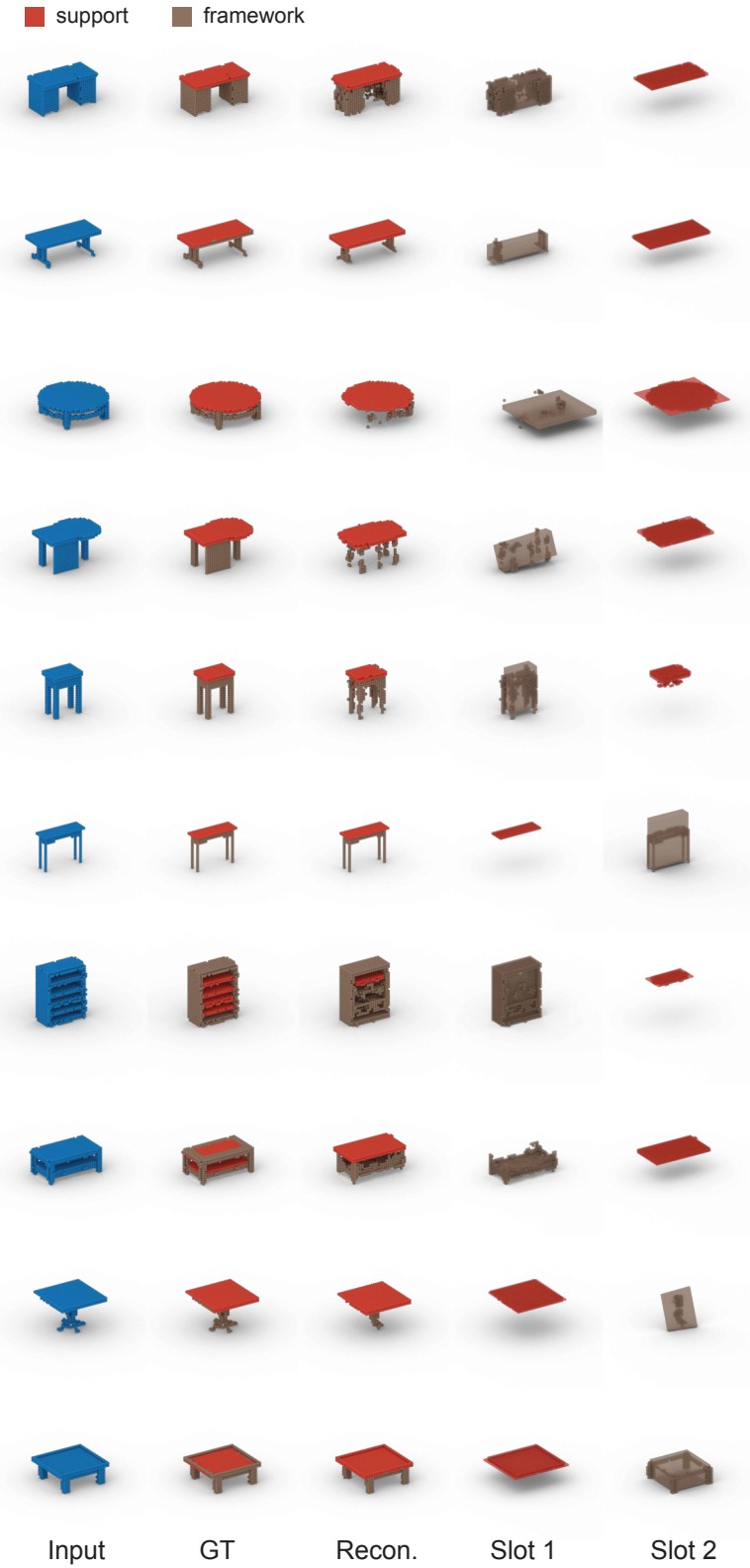

Figure 17: Additional qualitative results on objects in our "support" subset.

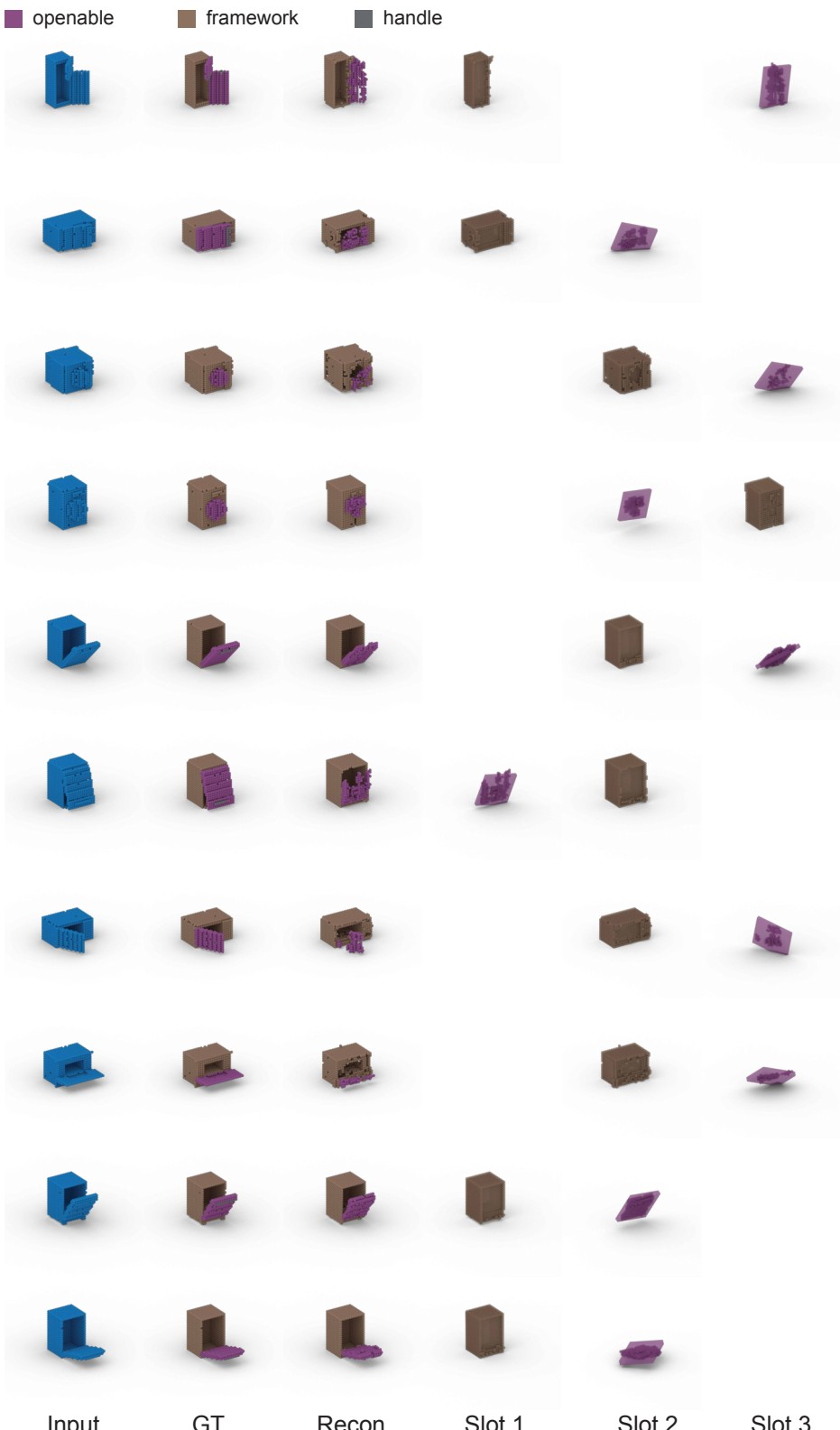

Figure 18: Additional qualitative results on objects in our "openable" subset.

