# OpenReview forum: "PartAfford: Part-level Affordance Discovery"
_ICLR.cc/2023/Conference — Submitted to ICLR 2023_

### Official Review · Reviewer_iNCK · 2022-10-23

**Confidence:** 4
**Clarity, Quality, Novelty And Reproducibility:** 1. The motivation for the work is not…
**Correctness:** 2
**Technical Novelty And Significance:** 2
**Empirical Novelty And Significance:** Not applicable
**Recommendation:** 3

**Strength And Weaknesses:**

Strength

1. The paper studies object affordance, a critical problem with various applications such as scene understanding, object understanding, and object manipulation. Toward the goal, a part-level affordance discovery task is proposed. Compared to the existing densely-labeled paradigm, the proposed task offers a new challenge to the community that studies affordance detection, weakly-supervised segmentation, and part segmentation. Note that the authors propose a part affordance dataset with 24 affordance categories shared among > 25, 000 objects.
2. Figure 5 demonstrates the effectiveness of the proposed components. In particular, the authors successfully demonstrate that the proposed task cannot be tackled simply with the slot-attention algorithm (Locatello et al., 2020). In addition, the effectiveness of the cuboidal primitive regularization branch can be observed in Figure 5 (c).

Weaknesses

1. Task formulation: The authors motivate the readers that the work aims to answer a question: what is the natural way to learn object affordance from geometry with humanlike sparse supervision? Then, the authors answer the question from a labeling perspective, i.e., use part-level affordance instead a densely-labeled strategy. It makes sense to reduce the annotation efforts by proposing the task. However, the reviewer is not convinced that the task tries to answer the question directly. Specifically, other affordance-related works (e.g., Kjellström et al., CVIU 2011 and Nagarajan et al., ICCV 2019, Wang et al., ECCV 2022) that study affordance from an interaction perspective. With a detailed discussion with them, the reviewer has a hard time appreciating the value of the work.
    1. Kjellström et al., Visual object-action recognition: Inferring object affordances from human demonstration, CVIU 115 (2011) 81-90
    2. Nagarajan et al., Grounded Human-Object Interaction Hotspots from Video, ICCV 2019
    3. Wang et al., “AdaAfford: Learning to Adapt Manipulation Affordance for 3D Articulated Objects via Few-shot Interactions,” ECCV 2022
2. Part affordance dataset: the authors did not recognize the existing dataset (i.e., Myers et al., 2015). Note that they also consider affordance detection in cluttered environments. The authors should discuss the difference and why the dataset Myers et al., 2015 built cannot serve the need for evaluation. It is worth noting that the number of affordance categories in the proposed dataset is larger than Myers et al., 2015. However, the authors only conduct experiments on seven affordances. The experiments weaken the advantage of the proposed dataset.
    1. Myers et al., "Affordance Detection of Tool Parts from Geometric Features," ICRA 2015
3. The proposed algorithm is highly relevant to 3D weakly supervised semantic segmentation with scene-level labels (Wei et al., CVPR 2020 and Ren et al., CVPR 2021). They could serve as the baseline for the proposed task. The authors should justify why these methods are not considered in the experiments. Please comment.
    1. Wei et al., Multi-Path Region Mining For Weakly Supervised 3D Semantic Segmentation on Point Clouds, CVPR 2020
    2. Ren et al., 3D Spatial Recognition without Spatially Labeled 3D, CVPR 2021
4. Limited quantitative evaluation: The part affordance dataset has various objects and affordance categories. However, the authors only report a subset (i.e., sittable, support, and openable) in Tables 1 and 2. In addition, the paper also reports similar qualitative analysis on similar objects. Currently, the experimental results are insufficient to justify the proposed method's effectiveness in a wide range of tasks.

**Summary Of The Paper:**

The work focuses on learning object affordance. The authors present a new task called part-level affordance discovery (PartAfford). Given only the affordance labels for each object, the machine is tasked to (i) decompose 3D shapes into parts and (ii) discover how each part of the object corresponds to a certain affordance category. The authors propose a slot-attention-based framework with 3D part reconstruction, affordance prediction, and cuboidal primitive regularization. They construct a part affordance dataset and benchmark their method with several baselines they designed.

**Summary Of The Review:**

Overall, the reviewer agrees that the proposed task is challenging. However, it is questionable to motivate the task from object affordance. In addition, the experimental section is not convincing. The reviewer looks forward to the author's feedback on the questions.

---

> ### Author Response · Authors · 2022-11-19
> **Response to Reviewer #4 iNCK**
>
>
> Thank you very much for your valuable feedback! We sincerely hope that our response can address your concerns.
>
> 1. Motivation/Task formulation: what is the natural way to learn object affordance?
>
>     > **Q:** The authors motivate the readers that the work aims to answer a question: what is the natural way to learn object affordance from geometry with humanlike sparse supervision? Then, the authors answer the question from a labeling perspective, i.e., use part-level affordance instead a densely-labeled strategy. It makes sense to reduce the annotation efforts by proposing the task. However, the reviewer is not convinced that the task tries to answer the question directly. Specifically, other affordance-related works (e.g., Kjellström et al., CVIU 2011 and Nagarajan et al., ICCV 2019, Wang et al., ECCV 2022) that study affordance from an interaction perspective. With a detailed discussion with them, the reviewer has a hard time appreciating the value of the work. The motivation for the work is not clear. Specifically, is the proposed task more humanlike to learning object affordance?
>
>     **A:** Thanks for sharing your opinion, and we think this could lead to an insightful discussion on affordance learning.
>
>     One important point we want to clarify first is that we are not arguing that "our proposed task is more humanlike to learn object affordance," but "in the regime of supervised learning, learning affordance from part-level affordance set is more humanlike compared with learning from dense annotations." From our perspective, affordance learning requires joint efforts from both supervised learning and interaction-based learning.
>
>     In the related work section, we discussed some prior arts that learn affordance from interaction, which can be divided into two streams. One stream learns 2D affordances from human demonstration videos, like Ego-Topo [4], which is a succeeding work of [2]. The other stream actively learns from simulation, like Where2Act [5], which is a follow-up work of [3]. Both streams, from the an interaction perspective, have their limitations.
>
>     On the one hand, work on video-based learning-from-demonstration usually infers high-level and coarse affordance information: [1] studies affordance in the context of action and object recognition, [4] associates affordance labels with zones from video frames, and [2] relies heavily on hands as probes to infer 2D heatmaps on images. The lack of grounding on 3D objects also poses limitations as they cannot directly facilitate real-world manipulation or motion planning.
>
>     On the other hand, active learning or simulation-based methods are often restricted to basic manipulations in specific domains. For example, only pulling and pushing are considered in [3] and [5]. Instead, our task does not rely on external demonstration video datasets or time-consuming simulations. We motivate researchers to explore 3D part-based affordance representation and reason about affordance across object geometry and category with less supervision. As noted in Section 6.5, conclusion, and Appendix C, we believe that future solutions to our task could involve those interaction cues to take the best of both worlds, further improving the performance and extending the scope. Our solution could also provide an intuitive prior for active learning methods, which could boost learning efficiency compared to learning from scratch.
>
>     We hope the additional explanation and discussion could help to clarify our motivation. We have included your suggested papers and further add these discussions in the revision.
>
>     [1] Kjellström et al., Visual object-action recognition: Inferring object affordances from human demonstration, CVIU 115 (2011) 81-90.
>
>     [2] Nagarajan et al., Grounded Human-Object Interaction Hotspots from Video, ICCV 2019.
>
>     [3] Wang et al., AdaAfford: Learning to Adapt Manipulation Affordance for 3D Articulated Objects via Few-shot Interactions. ECCV 2022.
>
>     [4] Nagarajan et al., Ego-topo: Environment affordances from egocentric video. CVPR 2020.
>
>     [5] Mo et al., Where2act: From pixels to actions for articulated 3d objects. ICCV, 2021.

---

> > ### Author Response · Authors · 2022-11-19
> > **Response to Reviewer #4 iNCK (Part 2)**
> >
> > 2. Discussion on the related work RGB-D Part affordance dataset (Myers et al., ICRA 2015)
> >
> >     > **Q:** the authors did not recognize the existing dataset (i.e., Myers et al., 2015). Note that they also consider affordance detection in cluttered environments. The authors should discuss the difference and why the dataset Myers et al., 2015 built cannot serve the need for evaluation.
> >
> >     **A:** Thanks for introducing a relevant paper, and we have cited it in the revision. The dataset in (Myers et al., ICRA 2015) provides pixel-level annotations in RGB-D images for 7 affordances of 105 tabletop tools. The RGB-D data cannot offer full 3D object structures due to occlusion and viewpoint, thus image cues are vital for approaches evaluated on this dataset. In contrast, our part affordance dataset contains complete 3D point clouds of over 25,000 object instances across 24 object categories with 24 kinds of affordance labels. Our method is also based on the assumption of known 3D object geometry, which cannot be offered by RGB-D data. So the domain gap between point cloud data and RGB-D data is the major reason why we cannot directly evaluate our method on this dataset.
> >
> > 3. Comparison with 3D weakly-supervised semantic segmentation algorithms.
> >
> >     > **Q:** The proposed algorithm is highly relevant to 3D weakly supervised semantic segmentation with scene-level labels (Wei et al., CVPR 2020 and Ren et al., CVPR 2021). They could serve as the baseline for the proposed task. The authors should justify why these methods are not considered in the experiments. Please comment.
> >     > 1. Wei et al., Multi-Path Region Mining For Weakly Supervised 3D Semantic Segmentation on Point Clouds, CVPR 2020.
> >     > 2. Ren et al., 3D Spatial Recognition without Spatially Labeled 3D, CVPR 2021.
> >
> >     **A:** Thanks for the pointer. 3D weakly supervised semantic segmentation with scene-level labels is a relevant topic, but related work is not considered in the experiment for the following reasons:
> >
> >     - 3D weakly supervised segmentation and affordance discovery are intrinsically different tasks. Although both utilize label sets as weak supervision, the 3D segmentation task focuses on separating different objects by **category-specific** features, whereas affordance discovery learns part-level **category-agnostic** features related to certain affordances ****across all the object categories.
> >     - 3D weakly supervised semantic segmentation methods often require additional inputs. For example, [1] takes both scene-level labels and subcloud-level labels for class region localization, and [2] needs additional geometric features of height and surface normal. Both [1] and [2] rely on color cues for segmentation, whereas our method only takes in voxelized 3D objects.
> >     - There is also a significant data domain gap between scene-level segmentation and affordance discovery. The segmentation granularity in 3D scenes is more sparse, where most objects keep their distance from each other. Algorithms designed for scene segmentation rely on this underlying assumption, so the [2] fails in "ambiguous objects" or when “multiple objects of the same classes are spatially close.” In contrast, 3D object parts are connected all together with no clear boundary in the affordance discovery, and the slot attention module in our proposed method encourages competence between slot features for better clustering and part abstraction.
> >
> >     The intrinsic task difference, special-designed technical facts, and data domain gap make the models on 3D weakly supervised semantic segmentation with scene-level labels unsuitable for the part-level object affordance discovery. We will add the discussion in the revision.
> >
> >     [1] Wei et al., Multi-Path Region Mining For Weakly Supervised 3D Semantic Segmentation on Point Clouds, CVPR 2020.
> >
> >     [2] Ren et al., 3D Spatial Recognition without Spatially Labeled 3D, CVPR 2021.

---

> > > ### Author Response · Authors · 2022-11-19
> > > **Response to Reviewer #4 iNCK (Part 3)**
> > >
> > > 4. Limited quantitative and qualitative evaluation.
> > >
> > >     > **Q:** It is worth noting that the number of affordance categories in the proposed dataset is larger than Myers et al., 2015. However, the authors only conduct experiments on seven affordances. The experiments weaken the advantage of the proposed dataset. The part affordance dataset has various objects and affordance categories. However, the authors only report a subset (i.e., sittable, support, and openable) in Tables 1 and 2. In addition, the paper also reports similar qualitative analysis on similar objects. Currently, the experimental results are insufficient to justify the proposed method's effectiveness in a wide range of tasks.
> > >
> > >     **A:** In the paper, we study three subsets related to the most representative affordance categories, “sittable,” “support,” and “openable” separately. These affordances are the most common cross-category affordances, e.g., “openable” in refrigerators, dishwashers, washing machines, and microwaves. The current three subsets also provide sufficient coverage of objects in the dataset, containing 7 kinds of affordances from 8 object categories and covering 17, 842/25K≈71% instances.
> > >
> > >     Here, we conduct further experiments and report performance on more affordance categories on the long tail of the dataset distribution. In Appendix A, we include the subsets “display” and “cutting.” The “display” set includes part-level affordances {display, wrapgrasp, pressable}  from 1,441 object instances, most of which are laptops and displays. The “cutting” set includes affordances {cutting, handle} from 641 object instances, most of which are knives and scissors. Now there are 11 kinds of affordances from 12 object categories involved in the experiments. Quantitative results are shown in the tables below. Our method still shows a significant advantage over the baseline methods on two core metrics: mIoU for part affordance discovery and MSE for 3D reconstruction. Qualitative results are shown in Fig. 10 and Fig. 11. We demonstrate that affordances like “display” could also be learned across categories. However, due to the lack of appearance information, our method finds it difficult to separate closely-connected parts (e.g., keys-keyboard, blade-handle). Scissor is the least common object category; thus, unsatisfactory results on them are also expected. As also discussed in Appendix C, affordance is naturally ambiguous; it is either especially challenging to segment or requires more than geometric information (e.g., active interactions) to discover the other affordance (e.g., “pressable” and “pinchable”). Our work provides a stepping stone towards learning complex affordance from a weak supervision set. Learning more affordance categories under unbalanced distribution and rare affordances beyond visual appearance will be a promising direction for future work.
> > >
> > >     Table. Quantitative results on "display".
> > >
> > >     | Method | mIoU(%) | MSE | AP(%) |
> > >     | --- | --- | --- | --- |
> > >     | Slot MLP | 17.91 | 0.0163 | 87.19 |
> > >     | Ours w/o Afford & Cuboid | 14.83 | 0.0146 | N/A |
> > >     | Ours w/o Cuboid | 46.97 | 0.0140 | 80.62 |
> > >     | Ours (full) | 47.56 | 0.0135 | 82.01 |
> > >
> > >     Table. Quantitative results on "cutting".
> > >
> > >     | Method | mIoU(%) | MSE | AP(%) |
> > >     | --- | --- | --- | --- |
> > >     | Slot MLP | 21.57 | 0.0030 | 97.67 |
> > >     | Ours w/o Afford & Cuboid | 27.75 | 0.0028 | N/A |
> > >     | Ours w/o Cuboid | 28.05 | 0.0027 | 96.89 |
> > >     | Ours (full) | 29.75 | 0.0026 | 97.67 |
> > >
> > >     As for the qualitative results, we have covered sufficient object variety in the PartNet dataset; see Fig. 4, 5, 6, and 7 in the main paper and Fig. 10, 11, 16, 17, and 18 in the Appendix. For example, for the "sittable" subset, we have included "park bench, armchair, office chair, stool, sofa, barber chair, cantilever chair, wing chair, deck chair, garden chair, etc" with various shapes and hierarchical structures.
> > >
> > >     To further demonstrate the generalizability of our methods qualitatively, we additionally provide evaluations on real scanned and more diverse objects from the Replica [1] dataset (Appendix E). Results are shown in Figure 8 and Figure 9 at the top of the Appendix.
> > >
> > >     Although the reconstructions may not be perfect, partly due to the reconstruction bottleneck’s impact on disentanglement quality (Engelcke et al., 2020), the learned model generalizes well and can identify the functional parts given novel real-world objects.
> > >
> > >     [1] Julian Straub, Thomas Whelan, Lingni Ma, Yufan Chen, Erik Wijmans, Simon Green, Jakob J Engel, Raul Mur-Artal, Carl Ren, Shobhit Verma, et al. The replica dataset: A digital replica of indoor spaces. *arXiv preprint arXiv:1906.05797*, 2019.

---

### Official Review · Reviewer_2e64 · 2022-10-23

**Confidence:** 3
**Correctness:** 3
**Technical Novelty And Significance:** 3
**Empirical Novelty And Significance:** 3
**Recommendation:** 6

**Clarity, Quality, Novelty And Reproducibility:**

Clarity, well, easy-to-follow

Quality, well, more figures would be better

Novelty, meaningful annotation upon PartNet

Reproducibility, code, and data are provided, but I did not check all of them in the supplementary carefully

**Strength And Weaknesses:**

Pros:
+ The aim and motivation of the benchmarks are sound and well-defined to make the community move on.

+ The dataset is large-scale and useful as a good complementary to PartNet.

+ The proposed method follows a sound design and uses slot attention as the feature bottleneck to finish three sub-tasks. And its performance looks well.

+ Code and data licenses are available.

Cons:
- The classes of affordance are limited. Even the dataset is large with a lot of objects. Furthermore, in tests, only several affordances are tested.

- Though there are analyses about the mutual effect of the three tasks, deeper insight is lacking to guide future study. I suggest a more profound discussion about the possible design and methods for future study, as this is a benchmark paper.

- Lacking a longtail distribution data analysis, e.g., affordances, object classes, parts, effect on performance, etc.

- More visualizations according to the above analysis would make this paper more solid.

- "We only keep the most prioritized affordance for each part to ease the ambiguities in learning", is somehow oversimplified.

- Possibilities scaling to some real scanned objects and more diverse objects?

- typo: in robotics (Nagarajan & Grauman, 2020; Mo et al., 2022).Prior --> ). Prior

**Summary Of The Paper:**

This paper proposes a large-scale 3D object affordance learning and part discovery dataset, paired with a baseline method. The proposed task is important for 3D object understanding and would be impactful in many fields. The dataset consists of more than 25,000 objects and each object has a set of affordance labels. The objects mainly come from PartNet and PartNet-Mobility. The task requires the machine to discover the parts of these human-made objects and classify their affordance under a weakly-supervised learning setting. The proposed method uses slot attention as the main component with a multi-task head design to address this task. In experiments, this method achieves decent performance compared with some basic methods.

**Summary Of The Review:**

Overall, this is an interesting paper with a lot of data, sound methods and experiment designs, and good writing. Though there is some room to improve (listed above), I think it is OK to be accepted.

---

> ### Author Response · Authors · 2022-11-19
> **Response to Reviewer #3 2e64**
>
> Thank you very much for your valuable feedback! We sincerely hope that our response can address your concerns.
>
> 1. Limited affordance classes and only several affordances are tested.
>
>     > **Q:** The classes of affordance are limited. Even the dataset is large with a lot of objects. Furthermore, in tests, only several affordances are tested.
>
>     **A:** Our PartAfford contains over 25, 000 3D CAD models from the PartNet dataset, and we provide annotations for 24 potential affordance labels with expert-defined descriptions to guarantee the variety and consistency of the mapping function. Details are shown in Section 5 and Appendix I. In the paper, we study three subsets related to the most representative affordance categories, “sittable,” “support,” and “openable” separately, where the subsets are created by collecting cross-category objects that share the corresponding affordance label in our dataset. These affordances are the most common cross-category affordances, e.g., “openable” in refrigerators, dishwashers, washing machines, and microwaves. The current three subsets also provide sufficient coverage of objects in the dataset, containing 7 kinds of affordances from 8 object categories and covering 17, 842/25K≈71% instances.
>
>     Here, we conduct further experiments and report performance on more affordance classes on the long tail of the dataset distribution. In Appendix A, we include the subsets “display” and “cutting.” The “display” set includes part-level affordances {display, wrapgrasp, pressable}  from 1,441 object instances, most of which are laptops and displays. The “cutting” set includes affordances {cutting, handle} from 641 object instances, most of which are knives and scissors. Now there are 11 kinds of affordances from 12 object categories involved in the experiments. Quantitative results are shown in the tables below. Our method still shows a significant advantage over the baseline methods on two core metrics: mIoU for part affordance discovery and MSE for 3D reconstruction. Qualitative results are shown in Fig. 10 and Fig. 11. We demonstrate that affordances like “display” could also be learned across categories. However, due to the lack of appearance information, our method finds it difficult to separate closely-connected parts (e.g., keys-keyboard, blade-handle). Scissor is the least common object category; thus, unsatisfactory results on them are also expected. As also discussed in Appendix C, affordance is naturally ambiguous; it is either especially challenging to segment or requires more than geometric information (e.g., active interactions) to discover the other affordance (e.g., “pressable” and “pinchable”). Our work provides a stepping stone towards learning complex affordance from a weak supervision set. Learning more affordance categories under unbalanced distribution and rare affordances beyond visual appearance will be a promising direction for future work.
>
>     Table. Quantitative results on "display".
>
>     | Method | mIoU(%) | MSE | AP(%) |
>     | --- | --- | --- | --- |
>     | Slot MLP | 17.91 | 0.0163 | 87.19 |
>     | Ours w/o Afford & Cuboid | 14.83 | 0.0146 | N/A |
>     | Ours w/o Cuboid | 46.97 | 0.0140 | 80.62 |
>     | Ours (full) | 47.56 | 0.0135 | 82.01 |
>
>     Table. Quantitative results on "cutting".
>
>     | Method | mIoU(%) | MSE | AP(%) |
>     | --- | --- | --- | --- |
>     | Slot MLP | 21.57 | 0.0030 | 97.67 |
>     | Ours w/o Afford & Cuboid | 27.75 | 0.0028 | N/A |
>     | Ours w/o Cuboid | 28.05 | 0.0027 | 96.89 |
>     | Ours (full) | 29.75 | 0.0026 | 97.67 |

---

> > ### Author Response · Authors · 2022-11-19
> > **Response to Reviewer #3 2e64 (Part 2)**
> >
> >
> > 2. More profound discussion about the possible design and future study.
> >
> >     **A:** We discussed some design insights, limitations, and potential future work in Section 6.5 Failure Cases and Appendix C: Further Analysis and Discussion. Below, we summarize and further enrich them as follows.
> >
> >     - The number of slots in the architecture: We set the number of slots as the maximal number of affordance labels that appear in one subset, which is different from the previous object-centric learning algorithm (Locatello et al., 2020), where the number of slots could be arbitrary. This is because when we increase the number of slots, we also increase the ambiguities of the affordance composition and set matching at the same time, which prevents the model from learning accurate correspondence between affordance labels and parts. Learning cross-category affordance and part correspondence with a fixed number of slots, especially with good slot initialization, will be a promising future direction to improve performance on the task.
> >     - Fine-grained object shape details: Our model faces difficulties in discovering and reconstructing fine-grained 3D shapes, especially tiny parts (e.g., “handle”), as the objects with related affordances come from various object categories with diverse shapes and detailed granularity, making it challenging for the model to capture such complex mixtures of distributions. This points out future directions to better understand object parts (e.g., segment, reconstruct) by improving the model’s capability and more carefully designed architecture.
> >     - Multiple affordances per part: Our dataset supports learning multiple affordances per part as we annotate a single part with multiple affordances (see details in Sec. 5). Yet, as the first step towards affordance discovery, our current method focuses on learning the most important affordance. As one of the future directions discussed in Appendix C.3, our current method is extendable to learning multiple affordances by switching the one-hot affordance label to multi-hot labels. Our ablation experiments also show that the single-hot affordance prediction appears significant to guide the slot competition, so we think fine-tuning the multi-hot affordance prediction based on the one-hot version may be a viable way for extension.
> >     - Learning richer affordances: The current experiment setting considers three subsets of PartAfford, as learning all the affordances together introduces significant ambiguities to part-affordance correspondences and challenges the 3D object-centric model. For some affordance categories (e.g., “pressable” and “pinchable”), It is either especially challenging to segment or requires more than geometric information (e.g., active interactions) to discover. Our work provides a stepping stone towards learning complex affordance from a weak supervision set. Learning more affordance categories under unbalanced distribution and rare affordances beyond visual appearance will be a promising direction for future work.
> >
> >     We have added this discussion in the revision.
> >
> > 3. Long-tail distribution data analysis and visualizations.
> >
> >     **A:** In Figure 12 (see Appendix), we provide statistical visualizations for the long-tail distribution of our dataset. Most of the affordance categories are annotated on more than 2 kinds of objects (Figure 12a). Our curated subsets cover the most common affordance categories (top 7), as well as some rare affordance categories (e.g., "cutting") on the long tail (Figure 12b).
> >
> > 4. Scaling to some real scanned objects and more diverse objects.
> >
> >     **A:** To further demonstrate the generalizability of our methods qualitatively, we additionally provide evaluations on real scanned and more diverse objects from the Replica [1] dataset. Results are shown in Figure 8 and Figure 9 at the top of the Appendix.
> >
> >     Although the reconstructions may not be perfect, partly due to the reconstruction bottleneck’s impact on disentanglement quality (Engelcke et al., 2020), the learned model generalizes well and can identify the functional parts given novel real-world objects.
> >
> >     [1] Julian Straub, Thomas Whelan, Lingni Ma, Yufan Chen, Erik Wijmans, Simon Green, Jakob J Engel, Raul Mur-Artal, Carl Ren, Shobhit Verma, et al. The replica dataset: A digital replica of indoor spaces. *arXiv preprint arXiv:1906.05797*, 2019.
> > 5. Typo.
> >
> >     **A:** Thanks a lot for pointing out the typo. It has been fixed in the revision.

---

### Official Review · Reviewer_Xneh · 2022-10-25

**Confidence:** 5
**Correctness:** 4
**Technical Novelty And Significance:** 4
**Empirical Novelty And Significance:** 4
**Recommendation:** 8

**Clarity, Quality, Novelty And Reproducibility:**

The paper reads well. The presentation quality is decent. Novelty is great. Code is provided and hence reproducibility is guaranteed.

**Strength And Weaknesses:**

Strengths
1. The proposed task is interesting yet challenging and has value for many research communities.
2. Code is provided.
3. The video is very clear.

Weaknesses
1. Given that this paper is really interesting, it is very important to have a section that talks about limitations and future work in other problem domains, e.g., robotics.
2. I would also suggest that in camera ready it would be valuable to create a website that has an interactive viewer for people to view the dataset.

**Summary Of The Paper:**

This paper studies the problem of discovering part affordances. This is an important topic in many applications. Previous methods focused on learning object affordances with dense supervision. In this work, the paper proposes a new task of part affordance discovery. This task is interesting and challenging The paper constructs a part-level, cross-category 3D object affordance dataset with 24 affordance categories shared among 25,000 objects.

**Summary Of The Review:**

Please see the comments in the two boxes above.

---

> ### Author Response · Authors · 2022-11-19
> **Response to Reviewer #2 Xneh**
>
> Thank you very much for your valuable feedback! We sincerely hope that our response can address your concerns.
>
> 1. Limitations and Future work.
>
>     > **Q:** *It is very important to have a section that talks about limitations and future work in other problem domains, e.g., robotics.*
>
>     **A:** We have discussed some limitations and potential future work in "Section 6.5 Failure Cases" and "Appendix C: Further Analysis and Discussion." Below, we further summarize and enrich the revision with potential impact on related fields.
>
>     Limitations and future work:
>
>     - Fine-grained object shape details: Our model faces difficulties in discovering and reconstructing fine-grained 3D shapes, especially tiny parts (e.g., “handle”), as the objects with related affordances come from various object categories with diverse shapes and detailed granularity, making it challenging for the model to capture such complex mixtures of distributions. This points out future directions to better understand object parts (e.g., segment, reconstruct) by improving the model’s capability and more carefully designed architecture.
>     - Multiple affordances per part: Our dataset supports learning multiple affordances per part as we annotate a single part with multiple affordances (see details in Sec. 5). Yet, as the first step towards affordance discovery, our current method focuses on learning the representative affordance per part. As one of the future directions discussed in Appendix C.3, our current method is extendable to learning multiple affordances by switching the one-hot affordance label to multi-hot labels. Our ablation experiments also show that the single-hot affordance prediction appears significant to guide the slot competition, so we think fine-tuning the multi-hot affordance prediction based on the one-hot version may be a viable way for extension.
>     - Learning richer affordances: The current experiment setting considers three subsets of PartAfford, as learning all the affordances together introduces significant ambiguities to part-affordance correspondences and challenges the 3D object-centric model. For some affordance categories (e.g., “pressable” and “pinchable”), It is either especially challenging to segment or requires more than geometric information (e.g., active interactions) to discover. Our work provides a stepping stone towards learning complex affordance from a weak supervision set. Learning more affordance categories under unbalanced distribution and rare affordances beyond visual appearance will be a promising direction for future work.
>
>     Potential impact: By learning to discover the part-level affordance, our model could facilitate related topics like holistic scene understanding, human-object interaction, object manipulation, etc. We discuss and show how conditional 3D human motion/human-object interaction synthesis may be enabled by object part affordance discovery in Appendix C.4. Additionally, our work could provide priors for robotic cross-category object manipulation and task planning [1, 2, 3]. For example, [4] demonstrates a single reference image of an object with annotated affordance regions can help the robot to use the affordance skill in the real-world setting, where our method can provide generalizable affordance information on the 3D shapes without the need for manual annotation.
>
>     [1] Marc, et al. "Differentiable Physics and Stable Modes for Tool-Use and Manipulation Planning-Extended Abtract." *IJCAI*. 2019.
>
>     [2] Reed, et al. "Pddlstream: Integrating symbolic planners and blackbox samplers via optimistic adaptive planning." *Proceedings of the International Conference on Automated Planning and Scheduling*. Vol. 30. 2020.
>
>     [3] Reed, et al. "Integrated task and motion planning." *Annual review of control, robotics, and autonomous systems* 4 (2021): 265-293.
>
>     [4] Denis, et al. "One-Shot Transfer of Affordance Regions? AffCorrs!." Proceedings of the 6th Conference on Robot Learning (CoRL) (2022).
>
> 2. Project website with dataset gallery and interactive viewer.
>
>     > **Q:** *In camera ready it would be valuable to create a website that has an interactive viewer for people to view the dataset.*
>
>     **A:** We are working to set up a project website with an interactive 3D dataset viewer to show a variety of objects with our annotations, as well as paper, code, video demo, etc.

---

> > ### Comment · Reviewer_Xneh · 2022-11-22
> > **Post rebuttal comments**
> >
> > I am satisfied with the author response and would like to remain my rating.

---

### Official Review · Reviewer_X6Ff · 2022-10-25

**Confidence:** 4
**Correctness:** 4
**Technical Novelty And Significance:** 3
**Empirical Novelty And Significance:** 3
**Recommendation:** 8

**Clarity, Quality, Novelty And Reproducibility:**

### Clarity
The paper is straightforward and easy to follow. Some editing suggestions:
- Figure 1 left: I think **sparse** is confusing here. How about "weak"?
- Figure 2 (d): the illustration of overlapping cuboids and parts is very confusing. It looks like that is another reconstruction while it is an intuitive illustration of the loss.

### Quality
The quality of the work is good.

### Novelty
The 3D part affordance dataset is novel. The proposed workflow contains components from prior work such as the 3D feature and position embedding and 3D slot attention mechanisms but the combination of everything to solve this new task is novel.

### Reproducibility
There seems to be a sufficient amount of detail in the paper for reproducibility. The author will release the code and data.

**Strength And Weaknesses:**

### Strength
- The new task, PartAfford, is interesting and I believe will promote new research in this direction.
- The new dataset is also a valuable contribution to the community. I especially like the openable affordance shapes (Figure 3c).
- The proposed method to discover 3D parts and associate affordance in a weakly supervised setting is interesting.
- I like Section 6.4 which evaluates the generalization power of the proposed method on unseen objects.


### Weakness
- Although the 3D part affordance dataset contains 24 affordance labels, the experiment section only considers three: 'sittable', 'support', and 'openable'. I understand that they are very useful functions for everyday objects but I wish the method is evaluated on more categories for a comprehensive understanding of the task.

- I know the paper proposes a new task. But I think it could compare its individual components to other methods mentioned in the related work. For example, (unsupervised) 3D part discovery work could be compared. Also, even a fully supervised dense affordance learning scheme can be compared as a reference too because it is common to have such a method in the evaluation as the performance upper bound reference.

- Section 6.1 mentions that the current setting only allows one affordance per part. I think quite frequently an object part can afford multiple functions. It is not clear to me if the method is able to handle multiple affordance per part.

- Comparing to Table 1, Table 2 doesn't have results of Slot MLP and IODINE. A complete table will be better.


**Summary Of The Paper:**

This paper proposes the new task of part-level affordance discovery: it is a joint task of decomposing 3D shapes into their parts and predicting how each part corresponds to affordances. A learning framework has been proposed for this task. It learns to segment 3D shapes into parts from weak shape-level labels. It also associates affordances with predicted parts. The framework is powered by 3D position-embedded features and 3D slot attention. To facilitate this new task, a novel dataset that features part-level cross-category 3D object affordances is constructed. Extensive evaluation and ablation studies have been performed on the dataset.

**Summary Of The Review:**

Overall, I like this paper as it defines an interesting yet challenging task, i.e., jointly finding 3D parts and their affordance, and provides a viable solution. The new 3D part affordance dataset is also a nice contribution; it will promote more interesting research in this direction. There are a few places the paper could be improved as mentioned in the "Weakness" above. But I think the paper will be a good contribution to ICLR 2023 in its current form.

---

> ### Author Response · Authors · 2022-11-19
> **Response to Reviewer #1 X6Ff**
>
> Thank you very much for your valuable feedback! We sincerely hope that our response can address your concerns.
>
> 1. Evaluation of more affordance categories in our part affordance dataset.
>
>     > **Q:** Experiments on more categories. > Although the 3D part affordance dataset contains 24 affordance labels, the experiment section only considers three: 'sittable', 'support', and 'openable'. I understand that they are very useful functions for everyday objects, but I wish the method is evaluated on more categories for a comprehensive understanding of the task.
>
>     **A:** In the paper, we study three subsets related to the most representative affordance categories, “sittable,” “support,” and “openable,” separately, where the subsets are created by collecting cross-category objects that share the corresponding affordance label in our dataset. These affordances are the most common cross-category affordances, e.g., “openable” in refrigerators, dishwashers, washing machines, and microwaves. The current three subsets also provide sufficient coverage of objects in the dataset, containing 7 kinds of affordances from 8 object categories and covering 17, 842/25K≈71% instances.
>
>     Here, we conduct further experiments and report performance on more affordance categories on the long tail of the dataset distribution. In Appendix A, we include the subsets “display” and “cutting.” The “display” set includes part-level affordances {display, wrapgrasp, pressable}  from 1,441 object instances, most of which are laptops and displays. The “cutting” set includes affordances {cutting, handle} from 641 object instances, most of which are knives and scissors. Now there are 11 kinds of affordances from 12 object categories involved in the experiments. Quantitative results are shown in the tables below. Our method still shows a significant advantage over the baseline methods on two core metrics: mIoU for part affordance discovery and MSE for 3D reconstruction. Qualitative results are shown in Fig. 10 and Fig. 11. We demonstrate that affordances like “display” could also be learned across categories. However, due to the lack of appearance information, our method finds it difficult to separate closely-connected parts (e.g., keys-keyboard, blade-handle). Scissor is the least common object category; thus, unsatisfactory results on them are also expected. As also discussed in Appendix C, affordance is naturally ambiguous; it is either especially challenging to segment or requires more than geometric information (e.g., active interactions) to discover the other affordance (e.g., “pressable” and “pinchable”). Our work provides a stepping stone towards learning complex affordance from a weak supervision set. Learning more affordance categories under unbalanced distribution and rare affordances beyond visual appearance will be a promising direction for future work.
>
>     Table. Quantitative results on "display".
>
>     | Method | mIoU(%) | MSE | AP(%) |
>     | --- | --- | --- | --- |
>     | Slot MLP | 17.91 | 0.0163 | 87.19 |
>     | Ours w/o Afford & Cuboid | 14.83 | 0.0146 | N/A |
>     | Ours w/o Cuboid | 46.97 | 0.0140 | 80.62 |
>     | Ours (full) | 47.56 | 0.0135 | 82.01 |
>
>     Table. Quantitative results on "cutting".
>
>     | Method | mIoU(%) | MSE | AP(%) |
>     | --- | --- | --- | --- |
>     | Slot MLP | 21.57 | 0.0030 | 97.67 |
>     | Ours w/o Afford & Cuboid | 27.75 | 0.0028 | N/A |
>     | Ours w/o Cuboid | 28.05 | 0.0027 | 96.89 |
>     | Ours (full) | 29.75 | 0.0026 | 97.67 |
>
> 2. Comparison with unsupervised part discovery and fully-supervised methods.
>
>     > **Q:** I know the paper proposes a new task. But I think it could compare its individual components to other methods mentioned in the related work. For example, (unsupervised) 3D part discovery work could be compared. Also, even a fully supervised dense affordance learning scheme can be compared as a reference too because it is common to have such a method in the evaluation as the performance upper bound reference.
>
>     **A:** In the experiment section, we adapt IODINE, a 2D object discovery method, as a baseline. To provide a performance upper bound reference, we compare a fully supervised dense affordance learning method based on PointCNN with our method; qualitative and quantitative results are summarized in Appendix H and Appendix Table 3, respectively. We summarize the comparison below, which shows a reasonable gap between weakly-supervised methods and fully-supervised methods.
>
>     | mIoU (%) | sittable | support | openable |
>     | --- | --- | --- | --- |
>     | Weak supervision | 57.3 | 52.7 | 47.6 |
>     | Full supervision | 74.9 | 83.7 | 66.3 |

---

> > ### Author Response · Authors · 2022-11-19
> > **Response to Reviewer #1 X6Ff (Part 2)**
> >
> > 3. Model extension to multiple affordances per part.
> >
> >     > **Q:** Section 6.1 mentions that the current setting only allows one affordance per part. I think quite frequently an object part can afford multiple functions. It is not clear to me if the method is able to handle multiple affordance per part.
> >
> >     **A:** Good observation! Our dataset has supported learning multiple affordances per part; we annotate a single part with multiple affordances (see details in Sec. 5). Yet, as the first step towards affordance discovery, our current method focuses on learning the most representative affordance per part. As one of the future directions, already discussed in Appendix C.3, our current method is extendable to learning multiple affordances by switching the one-hot affordance label to multi-hot labels.
> >
> >     We also suggest further fine-tuning the multi-hot affordance prediction based on the one-hot version, since the affordance prediction of single-hot labels appears significant to guide the slot competition from our ablation study. We have added this discussion in the revision.
> >
> > 4. Complete table 2 with more baseline results.
> >
> >     > **Q:** Comparing to Table 1, Table 2 doesn't have results of Slot MLP and IODINE. A complete table will be better.
> >
> >     **A:** We provide more comparisons here and also add them in the revision. The conclusion remains the same:
> >
> >     The proposed method achieves the best overall performance on the PartAfford task, especially in the part discovery (mean IoU) where it outperforms the baselines by a large margin. Slot MLP achieves the best affordance prediction performance (AP) on "openable" but fails in the part discovery (mean IoU) and 3D reconstruction (MSE). Slot MLP cannot segment the object input into parts due to the lack of abstraction capability.
> >
> >     Table. Quantitative results on "support".
> >
> >     | Method | mIoU(%) | MSE | AP(%) |
> >     | --- | --- | --- | --- |
> >     | Slot MLP | 36.8 | 0.0099 | 91.6 |
> >     | Ours w/o Afford & Cuboid | 34.8 | 0.0092 | N/A |
> >     | Ours w/o Cuboid | 51.3 | 0.0087 | 95.2 |
> >     | Ours (full) | 52.7 | 0.0085 | 95.1 |
> >
> >     Table. Quantitative results on "openable".
> >
> >     | Method | mIoU(%) | MSE | AP(%) |
> >     | --- | --- | --- | --- |
> >     | Slot MLP | 21.0 | 0.0130 | 70.8 |
> >     | Ours w/o Afford & Cuboid | 19.9 | 0.0104 | N/A |
> >     | Ours w/o Cuboid | 46.7 | 0.0097 | 55.8 |
> >     | Ours (full) | 47.6 | 0.0093 | 60.4 |
> > 5. Editing suggestions.
> >
> >     Thanks for your suggestions! We modify them accordingly in the revised manuscript. In Figure 1, “sparse” has been changed to “weak.” In Figure 2(d), we have added a sentence for clarification: Each predicted cuboid in (d) wraps around the corresponding predicted object part in (b) tightly.

---

### Author Response · Authors · 2022-11-19
**General Comments**

We thank all the reviewers for their time and valuable comments. We sincerely hope that our response can address your concerns. We summarize the revisions in the main paper and appendix as the reviewers suggest, shown in red.

- Conduct further experiments on more affordance categories on the long tail of the dataset distribution. More specifically, we include the subsets "display" and "cutting." Experiment details are included in Appendix A, quantitative results are shown in Appendix Table 3, and qualitative results are shown in Appendix Figures 10 and 11.
- Complete Table 2 by adding more baseline results.
- Demonstrate the generalizability of our methods qualitatively, by additionally providing inference examples for real scanned and more diverse objects from the Replica (Straub et al., 2019) dataset in Appendix E. Results show that our method can generalize well to in-the-wild objects with noisy scans (Figure 8 and Figure 9 at the top of the Appendix).
- Add discussion about methods on weakly supervised semantic segmentation with scene-level labels in Sec. 2 Related Work.
- Extend discussion on the interaction-based affordance learning in Sec. 2 Related Work, and how to potentially integrate them with our task in Appendix C.2.
- Extend the discussion on the limitation of a single affordance label per part, and offer possible ways for extension based on our insights from the ablation study in Appendix. C.3.
- Extend the discussion on the potential impact of our proposed task and dataset on related fields (e.g., robotics) in Appendix C.4.
- Add dataset statistics figures and analysis in Appendix Figure 12.
- Add additional citations to more related work from the reviews.
- Revise typos per editing suggestions in Figure 1, Figure 2, and Related Work.

---

### Decision · Program_Chairs · 2023-01-20

**Decision:**

Reject

**Justification For Why Not Higher Score:**



The reviewers raised concerns regarding the value of studying  affordance in this categorical labelling setting. Given that these labels, e.g., ``sittable”, do not provide any additional information on how to sit, what is the benefit of assigning this label in the first place, in place of chair seat, or car seat, or table, and then simply supply associate table, chair seat and car seat categories to sittable category.

The rebuttal clarifies that the main point of the paper is to alleviate dense labelling in favor of weak annotations. Then, it would be natural for the paper to compare against previous weakly supervised segmentation methods, and repurpose their model for their task. However the rebuttal explains why this repurposing is not possible.

The reviewers further raise concerns why not all labels are used for training affordance models, and the rebuttal shows experiments with expanded label sets. The rebuttal explains that  some labels have very few examples.
Overall, the rebuttal notes: “Our work provides a stepping stone towards learning complex affordance from a weak supervision set.” but the proposed formulation remains unconvincing and the weakly supervised method proposed  is disconnected to previous weakly supervised methods and it appears hard to connect or re-use in a different setting. Thus, the paper is not suggested for publication in its current form.

**Justification For Why Not Lower Score:**

N/A

**Metareview: Summary, Strengths And Weaknesses:**

The paper presents a new task called part-level affordance discovery (PartAfford). Given only the affordance labels for each object, the method is tasked to (i) decompose 3D shapes into parts and (ii) discover how each part of the object corresponds to a certain affordance category. The authors propose a slot-attention-based framework with 3D part reconstruction, affordance prediction, and cuboidal primitive regularization. They construct a part affordance dataset with 24 affordance categories shared among > 25, 000 objects and benchmark their method with several baselines they designed.

The reviewers raised concerns regarding the value of studying  affordance in this categorical labelling setting. Given that these labels, e.g., ``sittable”, do not provide any additional information on how to sit, what is the benefit of assigning this label in the first place, in place of chair seat, or car seat, or table, and then simply supply associate table, chair seat and car seat categories to sittable category.

The rebuttal clarifies that the main point of the paper is to alleviate dense labelling in favor of weak annotations. Then, it would be natural for the paper to compare against previous weakly supervised segmentation methods, and repurpose their model for their task. However the rebuttal explains why this repurposing is not possible.

The reviewers further raise concerns why not all labels are used for training affordance models, and the rebuttal shows experiments with expanded label sets. The rebuttal explains that  some labels have very few examples.
Overall, the rebuttal notes: “Our work provides a stepping stone towards learning complex affordance from a weak supervision set.” but the proposed formulation remains unconvincing and the weakly supervised method proposed  is disconnected to previous weakly supervised methods and it appears hard to connect or re-use in a different setting. Thus, the paper is not suggested for publication in its current form.




**Summary Of Ac-Reviewer Meeting:**

We agreed that the affordance definition the paper provides is not convincing and that the weakly supervised algorithm is disconnected and cannot be used anywhere else, as it is also not benchmarked in any other setting.